# Benchmarking Physical Reasoning of Video Generative Models with Real Physical Experiments on Newtonian Mechanics.

## Abstract

Recent advances in image and video generation raise hopes that these models possess world modeling capabilities—the ability to generate realistic, physically plausible videos. This could revolutionize applications in robotics, autonomous driving, and scientific simulation. However, before treating these models as world models, we must ask: Do they adhere to physical laws? Current evaluation methods rely on subjective judgments or trajectory matching, limiting their usage for physical reasoning estimation, where many generations could be physically plausible. Thus, we introduce **Morpheus**, a new benchmark for evaluating video generation models on physical reasoning. It features 130 real-world videos capturing physical phenomena, guided by conservation laws. Using those as conditioning for video generation, we assess physical plausibility using physics-informed metrics evaluated with respect to infallible conservation laws known per physical setting, leveraging advances in physics-informed neural networks and vision-language foundation models. Our findings reveal that even with advanced prompting and video conditioning, current models struggle to encode physical principles despite generating aesthetically pleasing videos.

## 1 Introduction

Video generative models (VGMs) such as SORA (Brooks et al., 2024a), COSMOS (Agarwal et al., 2025), and Veo3 (Veo-Team et al., 2024) have taken the world by storm, building upon remarkable advances in image generative models (Ramesh et al., 2021; Saharia et al., 2022; Podell et al., 2023; Yu et al., 2023), and achieving unprecedented levels of visual fidelity and realism. These developments have not only pushed the boundaries of visual aesthetics but have also inspired the community to envision video generative models as potential *world models* (Cho et al., 2024; Agarwal et al., 2025). A world model, in this context, is more than just a system for generating frames, however; it is a model capable of understanding and predicting the dynamics, causal interactions, and underlying mechanisms of the physical world. Accurately benchmarking the physical dynamics of video generation is a critical requirement —and the focus of this work— toward adopting them potentially as world models.

Perhaps the most daring challenge from an AI perspective is the need for physical dynamics evaluation that goes beyond visual verification. Current methods use either a) human or VLM judgement (e.g., *"does this video of an object falling look legitimate?"*) (Bansal et al., 2024; Meng et al., 2024), or b) visual or geometric plausibility (e.g., *"is the generated scene visually consistent through time?"*) (Agarwal et al., 2025) or c) comparison with one possible future using trajectory matching (e.g., comparing the generated locations of a projectile with one possible ground-truth locations) (Kang et al., 2024; Motamed et al., 2025)). While human-based evaluations (potentially distilled in VLMs) are useful for initial screening of generated videos, they do not provide sufficient quantitative and objective evidence of physical plausibility. Trajectory matching evaluation can yield false negatives in cases where the generated video is physically plausible, but the object's trajectory deviates from a particular ground-truth trajectory due to unobserved initial conditions or other visually hidden factors, such as an object's mass or friction, that affect the object's motion.

A deeper physical understanding requires assessing whether a generated video preserves *physical invariants* and *physical dynamics* that govern the underlying system. For instance, in many systems,

Figure 1: Comparison of evaluation methods for video generative models. a) Human or VLM-based judgments provide only qualitative and subjective assessments of physical plausibility. b) Trajectory matching compares generated and ground-truth paths but may misclassify physically valid trajectories. For example, for projectile motion, many parabolic trajectories are physically plausible when VGMs are conditioned only on an image, as object velocity cannot be estimated from it. c) Our proposed framework, Morpheus, evaluates generated videos via physics-informed scores, testing both conservation of physical invariants and consistency with governing equations of motion.

quantities such as total energy must remain constant as the system evolves, providing opportunities for quantitative benchmarks of physical plausibility. In addition, we can test whether an object's trajectory is consistent with the set of trajectories permitted by the governing physical laws by evaluating the fit of the observed dynamics to the governing *equations of motion*. Using these quantitative evaluations, we can design systematic benchmarks that reveal whether video generative models (VGMs) truly capture the dynamics of the physical world or simply produce visually plausible approximations.

We propose **Morpheus**, a novel physics-informed benchmarking framework designed to evaluate the physical reasoning capabilities of video generative models using real-world physical experiments. The key idea behind **Morpheus** is to map video recordings of physical events—whether generated by models or recorded from real experiments—into a common physical representation that can be analyzed and compared. Leveraging advances in zero-shot object segmentation, object tracking, and physics-informed neural networks (PINNs) (Cuomo et al., 2022; He et al., 2023), our framework a) fits to the video dynamics the ODE that governs the underlying system, and b) extracts standardized physical measurements, such as velocity and acceleration from video data, which should conform to conservation laws. By collecting measurements from both real physical videos and generated ones, and comparing their summary statistics with respect to governing ODEs and physical invariants, **Morpheus** enables fair and systematic benchmarking of physical invariants, such as the conservation of energy or momentum, without requiring or relying on explicit ground truth data.

This work makes three contributions toward rigorous evaluation of physical reasoning in video generative models. First, we introduce **Morpheus**, the first benchmark to systematically evaluate physical reasoning based explicitly on physical invariants (Section 3) using real-world physical experiments to ground VGM's generations in a controllable setting. Second, we propose a novel framework that combines physics-informed deep learning with advanced computer vision techniques to enable coarse- and fine-grained analysis of physical phenomena (Section 4). Third, we evaluate state-of-the-art video generative models on **Morpheus** generating over 9000 videos, including CogVideoX (Yang et al., 2024b), PyramidalFlow (Jin et al., 2024), LTX-Video (HaCohen et al., 2024), Wan2.1 (Wan et al., 2025b), COSMOS (Agarwal et al., 2025) and Veo3 (Veo-Team et al., 2024), and show that while the best of these models excel in visual aesthetics, they fall short in modeling real-world physical dynamics (Section 5).

## 2 RELATED WORK

**Evaluation of VGMs.** Benchmarking video generation models have evolved to include comprehensive evaluation frameworks that assess multiple dimensions of video quality, temporal coherence, and alignment with prompts. Approaches like EvalCrafter (Liu et al., 2024a), VBench (Huang et al., 2024a), VBench++ (Huang et al., 2024b), AIGCBench (Fan et al., 2024), and TC-Bench (Feng et al., 2024) emphasize diverse metrics to evaluate visual fidelity, motion smoothness, spatial consistency, and temporal dynamics. For example, EvalCrafter (Liu et al., 2024a) uses metrics like Motion-Aware Consistency (MAC) and Scene Change Consistency (SCC) to assess the smoothness and natural

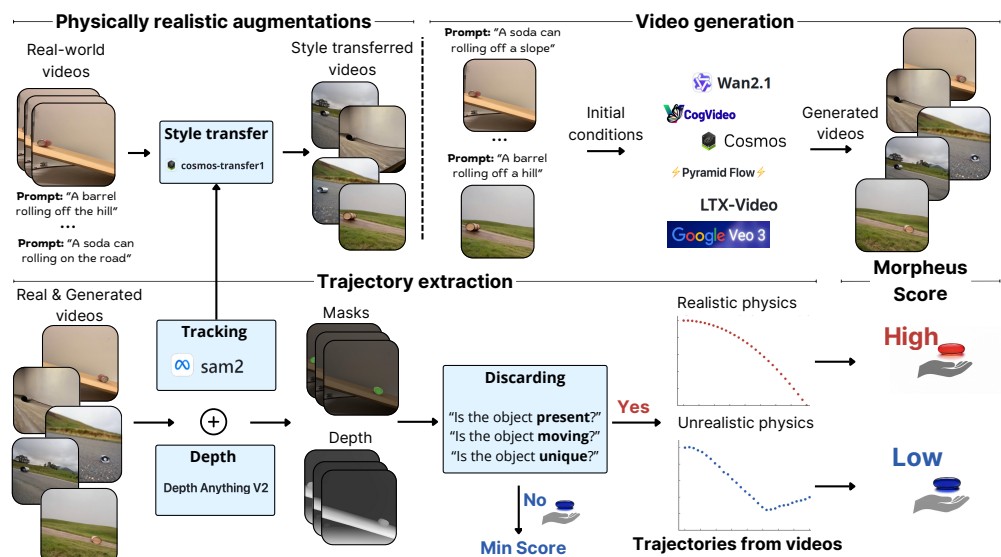

Figure 2: The overview of **Morpheus** benchmark. Video augmentation and generation (upper) and the trajectory extraction pipelines (lower). We start with augmenting recorded videos with realistic style transfer, based on object masks. Next, we use the first frame (or multiple frames in case of video conditioning) of the obtained videos, as well as the textual description, as a prompt for a VGM. After this, we extract object trajectories for both real-world and generated videos using the trajectory extraction pipeline, including trajectory tracking and discarding unreliable trajectories. Finally, we evaluate **Morpheus** scores for all videos with valid trajectories.

progression of motion, while VBench introduces metrics for spatial relationships and subject identity consistency to evaluate logical scene composition. Despite the breadth of these benchmarks, they primarily concentrate on perceptual and semantic aspects of video generation, whereas **Morpheus** focuses on physical plausibility of the generated videos.

**Physical reasoning and plausibility in VGMs.** Recent advances in evaluating physical plausibility in video generation have employed both human assessments (Bansal et al., 2024) and automated approaches leveraging vision-language models (VLMs) (Bansal et al., 2024; Meng et al., 2024) as well as object tracking metrics (Wang et al., 2024b; Motamed et al., 2025; Agarwal et al., 2025) (see Table 2 for comparison). Notable frameworks include VideoCon-Physics (Bansal et al., 2024), PhyGenBench (Meng et al., 2024) and PhysBench (Chow et al., 2025), which utilize VLMs to assess adherence to physical law prompts; VAMP (Wang et al., 2024b), which quantifies motion characteristics through acceleration and velocity variance; and Physics-IQ (Motamed et al., 2025) and COSMOS (Agarwal et al., 2025), which compare object masks between generated and real-world videos. Kang et al. (Kang et al., 2024) used the PHYRE simulator (Bakhtin et al., 2019) to fine-tune VGM on synthetic 2D data, facilitating out-of-distribution and combinatorial generalization evaluation. While frameworks like WISA (Wang et al., 2025) and T2VPhysBench (Guo et al., 2025) have underlined the importance of creating text-video caption pairs for testing the current capabilities of VGMs, they have one crucial limitation. Their application is limited to observing and providing scores to video related to human preference of *intuitive* (scores like Semantic Adherence and Physical Commonsense as defined in (Bansal et al., 2024), which is sufficient if VGMs are only used for the content creation and entrainment. Yet this approach is not enough for the objective evaluation of the physical plausibility of VGMs as the world models in robot learning.

Despite addressing diverse physical phenomena, these benchmarks have significant limitations. VLM and human evaluations often identify physical deviations categorically, like noting gravity violations without quantifying them. Moreover, VLM can hallucinate (Li et al., 2023) and miss subtle physical inconsistencies (Chow et al., 2025). On the other side, object tracking metrics are often based on simulated data (Agarwal et al., 2025; Bakhtin et al., 2019) and assume that modeled processes should be deterministic and predictable (Kang et al., 2024; Motamed et al., 2025). These limitations

Figure 3: Examples of physical experiments included in the **Morpheus** benchmark, illustrating both different dynamics and variations in object types. Top row (left to right): falling ball, projectile motion, holonomic pendulum, sliding, falling apple, and double pendulum. Bottom row (left to right): collision, falling tape, rolling can, spring, rolling orange, and bouncing.

highlight the critical need for more robust, interpretable benchmarks that can quantitatively evaluate physical realism by precisely measuring how well-generated videos preserve physical invariants and adhere to specific physical laws.

**Learn physical invariants and equations from data.** There is progress for learning conservation laws from trajectories (Liu & Tegmark, 2021), and equation discovery in hybrid dynamic systems (Liu et al., 2024b). Mechanistic Neural Networks (MechNN) (Pervez et al., 2024) are able to learn governing ODEs from data, while Mechanistic PDE Networks (Pervez et al., 2025) can learn Partial Differential Equations (PDEs). On the other hand, to compare the theoretical prediction with input data, PINNs (Cuomo et al., 2022; He et al., 2023), which integrate physical equations in the loss function, help identify possible physical factors causing errors (such as unmodeled friction, air drag, etc.) because it is able to learn corrections to make the predictions closer to the actual observed values.

## 3 MORPHEUS BENCHMARK

To rigorously examine discrepancies in adherence to physical laws within generated videos, we propose the **Morpheus** benchmark, consisting of a dataset for controlled conditioning, evaluation scores, and an analysis of state-of-the-art models' performance.

**Dataset methodology** We created a dataset of real-world videos of specific physical phenomena, focusing on capturing fundamental aspects of Newtonian mechanics. Videos were recorded under controlled laboratory conditions, allowing us to systematically vary initial parameters and capture repeatable scenarios. By operating in this rigorously controlled setting, we can isolate and test adherence to specific physical laws – such as the periodic dynamics of a harmonic pendulum – rather than merely assessing overall visual plausibility. This sets our dataset apart from previous works that often focus on uncontrolled, general-purpose video data (Motamed et al., 2025; Kang et al., 2024), allowing for a more precise and targeted evaluation of physical consistency.

We recorded a set of nine core physical experiments with 10-20 videos in each experiment, each highlighting different physical principles such as gravity (falling, projectile motion, and bouncing), periodic movements (spring and holonomic pendulum), friction and normal forces (incline sliding and rolling), and more complex dynamical systems such as multi-body collisions and double pendulum (See Fig. 3 and App. A for dataset an illustration and description). For each experiment, we varied the initial conditions (recording 5–7 videos per condition), such as speed for falling, angle and speed for projectile motion, and angle for pendulums. This diverse collection ensures robust coverage of dynamic behaviors, enabling thorough evaluation of generated videos against real-world physical phenomena. The initial video frame(s) are used as conditioning to guide VGM's sampling/generation process, ensuring the generated sequences start from the same conditions as the real experiments.

**Visually diverse conditioning for robust VGMs evaluation.** To obtain more robust evaluations, it is important to study the performance of VGMs under diverse natural initializations of the same physical process. This reduces sensitivity to the specific setup of the originally recorded experiments. Thus,

we augment the initial videos with visually diverse yet reliably generated variants. In particular, we use the video-to-video transfer method (Alhaija et al., 2025) with object masks extracted from recorded videos to obtain diverse and visually realistic videos following an additional text prompt for adaptation. While changing the semantics and appearance of the objects, the generated videos strictly follow the provided object masks and are reliable augmentations for conditioning in image-to-video and video-to-video generation. To ensure their quality, we filter augmented initializations through manual inspection for visual and physical plausibility. Thus, the original initializations are expanded 10-fold by generating 3 variations with 3 stylization prompts. For more details, we refer the reader to App. B.

**Metrics validation.** We use real-world videos as a *"gold standard"* to validate that our evaluation metrics work as intended. By analyzing the metrics on these ground-truth recordings, we demonstrate the reliability of our approach and establish an upper bound on performance, representing the precision with which we measure adherence to physical laws. In essence, the metrics computed on real-world videos provide a baseline for how closely any generative model can align with physical principles.

To structurally analyze the videos, we extract the trajectories of the objects by applying promptable video segmentation. These trajectories comprise 2D coordinates of the recognized objects through time and are used for further analysis with our physical metrics (see Sec. 3.2 for details).

## 3.1 PROMPTING METHODS

The physical dynamics of a scene are determined by its initial conditions: object positions, velocities, and geometric constraints (e.g., shapes, rigid connections). In generative models, these conditions are set through prompting, which can take three forms: *a)* textual prompts, *b)* single-image prompts, and *c)* video (multi-frame) prompts, providing a different level of control over generation. *Textual prompts* offer only broad control, suggesting behaviors (e.g., rolling, falling) without precise states. *Single-image prompts* improve precision by fixing initial locations, but lack motion detail. Only *video prompts* specify both positions and velocities, providing the highest control.

With this gradation in mind, we investigate how different levels of control affect the physical realism. We explore both textual prompt enhancement and various multi-frame prompting for models capable of leveraging these features (e.g. (Agarwal et al., 2025; Yang et al., 2024b)), allowing us to examine the relationship between input precision and output physical fidelity. Following (Yang et al., 2024b), we use a VLM (Zeng et al., 2024) to expand simple scene descriptions into richer prompts via instruction templates in zero- or few-shot settings. As not all VGMs provide their own prompt upsampler, we rely on the ChatGLM family of models (Zeng et al., 2024), while for COSMOS-variants (Agarwal et al., 2025) and WAN-2.1 (Wan et al., 2025a) we use their own devised (NVIDIA, 2024; Agrawal et al., 2024; Wang et al., 2024a) upsamplers respectively. In our evaluation, we create descriptive prompts with an emphasis on physical motion, and the upsampler brings the inference-time prompt distribution closer to that used during training.

## 3.2 TRAJECTORY EXTRACTION

While generated videos could be directly evaluated in terms of 3D consistency (Liu et al., 2024a) or other pixel-level generation properties (Agarwal et al., 2025), such evaluations are limited to visual and geometric realism of the generated videos. Instead, we are interested in how well these videos conform to physical laws. This means that we need to extract the relevant physical state variables, such as positions of objects, velocities, accelerations, masses, and so on. Thus, it is essential to transform the generated videos into perfect state variables of the depicted objects and their trajectories, which can then be further analyzed.

As we need to track objects in both real-world and generated videos, Segment Anything 2 (SAM-2) (Ravi et al., 2024) serve as a reliable way to generate 2D masks for any type of objects. We annotate the first frame of our videos in the dataset with positive and negative labels. Given the masks generated by SAM-2's we extarct the centroid of the object(s) (center-mass) in the video, at each frame of the video. In addition, we employ Depth Anything V2 (Yang et al., 2024a) to verify that objects have consistent depth through the sequence of movement (See physical score pipeline App. D.2). The depth values are calculated using the correspoding mask generated by SAM-2.

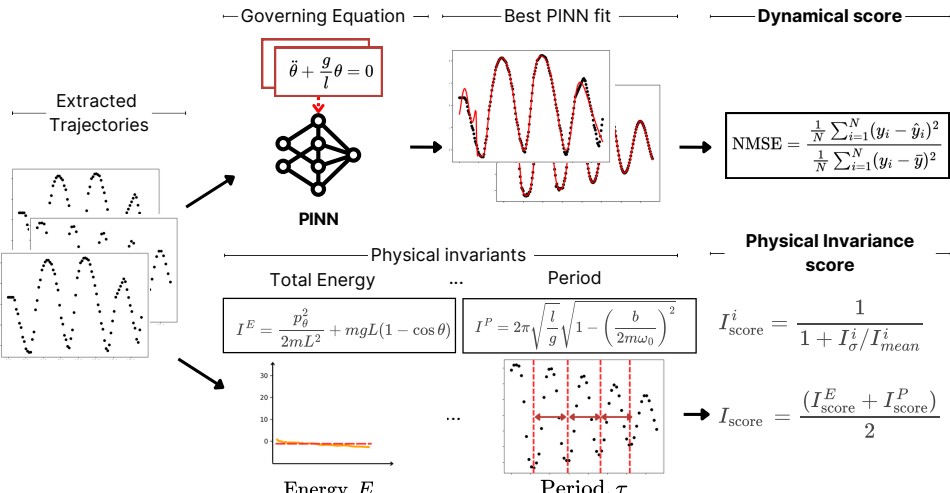

Figure 4: Evaluation of trajectories extracted from real and VGMs videos using our Dynamical (upper) and Physical Invariance (lower) scores. For the Dynamical score, trajectories from real-world or generated videos are fitted to a PINN with the corresponding equation of motion for the particular physical law as an extra loss term. For the Physical Invariance score, using the same trajectories, we estimate physical quantities that should be invariant in the systems, such as total energy and oscillation period, and use their variance as a measure of physical plausibility.

For velocity, acceleration and angular velocity, we employ the *central difference method* (Swanson & Turkel, 1992). To further reduce the noise, generated by the imperfections in the tracking pipeline, we follow with a series of smoothing operations, such as learning a linear regression with a sliding window and applying the Savitzky-Golay smoothing. The details can be found in the App. C.

## 4 PHYSICS-INFORMED EVALUATION METRICS

To assess the alignment of the generated video trajectories with physical laws, we propose a hierarchical evaluation framework for analyzing physical experiments in both real-world and generated videos.

**Discard rate** As a first metric, we compute the *discard rate*, which reflects the proportion of model-generated samples that must be discarded to ensure reliable trajectory extraction needed for Physical Invariances and Dynamical scores. The discard filtering is automatic and consists of three criteria: First, we discard generated videos where objects lack sufficient permanence throughout the video. Second, we discard generated videos which do not have a consistent number of objects. Finally, we discard generated videos if there is little motion detected, as such videos are not suitable for physical analysis. The overall discard rate represents the proportion of generated videos that fail at least one of these criteria. In addition, we verify that none of the real-world extracted trajectories are discarded, showing that our discard criteria are effective in distinguishing physical from non-physical videos. We provide further details on our filtering methodology in App. D.1. For the videos that pass filtering, we employ two metrics: *Dynamical score*, which measures adherence to the governing equation of motion, and *Physical Invariance score*, which quantifies invariance of conserved quantities such as energy or angular momentum (see Table 3 for all invariances).

**Physics beyond trajectory matching** Previous benchmarks such as PhysicsIQ (Motamed et al., 2025) and COSMOS (Alhaija et al., 2025) use trajectory matching by comparing generated trajectories with the ground-truth trajectories. As ground-truth trajectories themselves vary due to noise and hidden physical parameters that are not fully observable from visual image/video conditioning (e.g., object mass or friction), PhysicsIQ proposed to compare the obtained trajectory matching score with the variance in the real-world trajectories. However, such variance depends on the studied /recorded variations and can be arbitrarily large in cases where hidden parameters vary significantly.

Disappearance
Duplicates
Stillness

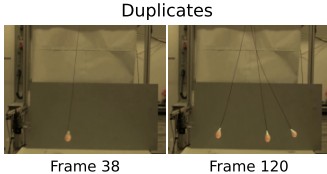
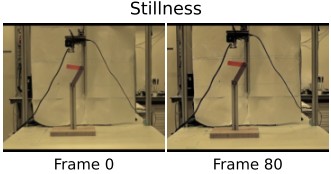

Frame 0      Frame 9      Frame 38      Frame 120      Frame 0      Frame 80

Figure 5: Different types of discarded generated videos: (left) A video showing the disappearance of the orange ball during fall; (middle) A video illustrating generation of multiple objects in pendulum experiment; (right) A video in which the double pendulum does not move.

Given the visual nature of the conditioning in VGMs (i.e., initial image or video), there are often parameters of the physical system that cannot be estimated from the provided *visual* conditioning. In the image conditioning case, such unobserved parameters exist for almost all conditions/experiments, limiting the trajectory matching metric for state-of-the-art image-based VGM, such as Veo3 (Veo-Team et al., 2024) and WAN2.1 (Wan et al., 2025a), which allow only image-based conditioning. Moreover, even in the video conditioning case there are often hidden parameters not observed in the original conditioning, limiting the applicability of trajectory matching only to simple systems. In contrast, **Morpheus** scores overcome those limitations by evaluating results of video generation on the level of physical laws instead of a particular trajectory provided by ground-truth videos.

## 4.1 DYNAMICAL SCORE

To calculate the Dynamical score, we use physics-informed neural networks (PINNs) (Cuomo et al., 2022), which directly incorporate physical laws as a prior. This setting allows us to learn the physical trajectory that fits the data the most, independent of the initial conditions. Fig. 4 illustrates our approach. A PINN is a neural network that receives a timestep $i$ of the trajectory as input and outputs the trajectory coordinates $\hat{T}_i$, velocity $\dot{\hat{T}}_i$, and acceleration $\ddot{\hat{T}}_i$. The model is typically trained with a loss function, comprising two components $L_{\text{data}}$ and $L_{\text{physics}}$: $L_{\text{total}} = L_{\text{data}} + \lambda L_{\text{physics}}$, where the $L_{\text{data}}$ is responsible for fitting the model to the datapoints, $L_{\text{data}} = \frac{1}{N} \sum_{i=1}^{N} \|\hat{T}_i - T_i\|^2$, and $L_{\text{physics}}$ enforces following the physical law. For each experiment, we explicitly implement the equation of motion in the form of an ordinary differential equation as PINN loss functions. E.g., for the falling ball, the equations of motion are: $\dot{x} = 0;$ $\ddot{y} + g = 0$, where $y$ is the vertical position, $\ddot{y}$ is the acceleration and $g$ is the gravitational constant. The $L_{\text{physics}}$ is calculated as: $L_{\text{physics}} = \frac{1}{M} \sum_{j=1}^{M} \left\| \ddot{\hat{y}}_j + g \right\|^2 + \left\| \ddot{\hat{x}} \right\|^2$, where $\ddot{\hat{y}}_j$ is the predicted acceleration derived from the PINN at the $j$-th time step.

**Computing the Dynamical score.** We use PINNs to assess the physical plausibility of trajectories from generated videos by computing the normalized mean square error (NMSE) of the model-learned trajectory derived from videos. We normalize by inverting the error, so 1 marks the best dynamical score and 0 a worse-than-constant PINN fit. A higher Dynamical score implies higher physical plausibility. For more details, please see App. D.5.

## 4.2 PHYSICAL INVARIANCE SCORE

To calculate a more fine-grained Physical Invariance score, we accompany each of our experiments with a list of physical invariances, i.e. values that we can derive from trajectories that stay constant in time. We make a series of reasonable assumptions about the setting and test them on the real-world trajectories. As invariances vary for different experiments, we present here one case study for the falling ball experiments, while describing all the other settings in App. D.6 and Table 3.

**Case study: Falling ball.** In the falling ball experiments, we have the following physical invariants.

$\Rightarrow$ *Total energy.* Assuming negligible air resistance, the total energy —the sum of the kinetic and potential energy— of the ball is conserved. The kinetic energy of the ball is: $T = \frac{1}{2}m(v_x^2 + v_y^2)$, where $v = (v_x, v_y)$ is the speed of the ball and $m$ is it's mass. Also, the potential energy is $V = mgy$ where $g$ is the gravitational acceleration constant and $y$ is the vertical coordinate. So, as the total energy is the sum of kinetic and potential, we get: $E = T + V = \frac{1}{2}m(v_x^2 + v_y^2) + mgy$.

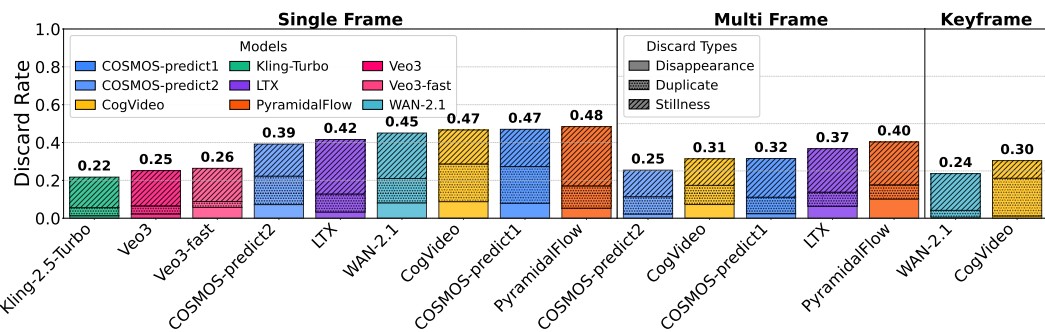

Figure 6: Average discard rates across all physical experiments (lower is better).

$\Rightarrow$ *Energy-to-mass ratio.* Assuming that the mass of the ball is constant, we derive the following invariant: $\frac{E}{m} = \frac{1}{2}(v_x^2 + v_y^2) + gy = $ const, which we can estimate with the data from our trajectory.

$\Rightarrow$ *Acceleration.* As no external forces are acting on the ball except for gravity, which is uniform and is directed downwards, the acceleration of the ball is also constant: $a_y = g = $ const.

$\Rightarrow$ *Horizontal momentum-to-mass ratio.* As with acceleration, the horizontal momentum, $p_x = mV_x$, is also preserved given no external forces. Thus, the horizontal velocity is conserved: $v_x = $ const.

**Computing the Physical Invariance score.** To convert the invariant into an actual score, like the Energy score, we calculate the standard deviation of the invariant time series and normalize it into the range of $(0, 1)$, with 1 indicating a perfect Physical Invariance score. As invariants must be by nature constant, a high standard deviation of these invariants (and thus a lower physical invariance score), indicates poor modeling of the respective physical invariants. In addition, for discarded trajectories we assign minimal Physical Invariance score equal to $0$. A detailed score calculation procedure is described in the App. D.7. For the derivations of each invariant we used, please refer to the App. D.6.

## 5 ANALYSIS

In this section, we analyze the results of the experiments. We present aggregated results for each model and conditioning type (single frame, multi-frame, or key frame interpolation) in Fig. 6 and Fig. 7. For each model, we average the results across all experiments. We select the best scores between using the *enhanced* (upgraded description, see Sec. 3.1) and *plain* (simple) versions of the textual prompt to represent the best model's ability.

Real-world videos consistently deliver optimal results across all experiments, as demonstrated by their minimal discard rates, high Dynamical scores ($0.96 - 0.99$), and consistently high Physical Invariance scores (above $0.90$). These metrics confirm the reliability of real-world videos as benchmarks for physically accurate and realistic motion, and validate the correctness of our experimental setups, providing the upper boundary for the performance of the video generation models.

Enhanced prompts typically improve performance metrics compared to plain prompts (see Fig. 24), although this trend varies depending on the specific model. Enhanced prompting leads to higher physical invariance and dynamical scores and lower discard rates in many cases (e.g., COSMOS-predict2 and CogVideo), yet occasionally decreases performance in models such as COSMOS-predict1, indicating that prompt enhancement effectiveness is context-dependent.

Multi-frame prompting and key frame interpolation generally outperform single-frame prompting for comparable models, achieving lower discard rates and higher dynamical and physical invariance scores, thereby demonstrating the advantage of increased temporal context. The prompting with first and last frames, performs notably well in specific experiments (e.g., holonomic pendulum with WAN-2.1, see Fig. 25), suggesting a promising direction for improving temporal coherence and physical realism, though limiting the ability to generate from scratch.

Close-sourced Kling-turbo, Veo3 and Veo3-fast model demonstrate good results, leading in single-frame conditioning with the scores 0.6, 0.56 and 0.53 correspondingly. Unfortunately, recent closed-

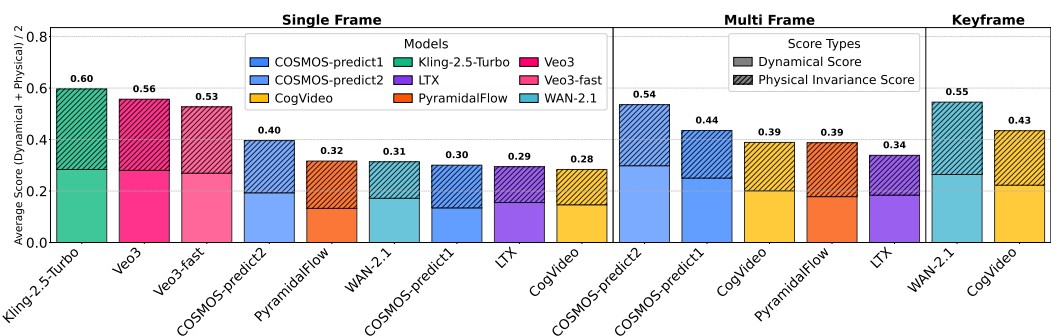

Figure 7: Aggregated scores across all physical experiments (higher is better).

source models like Veo do not support multi-frame conditioning; thus, we cannot draw any conclusion about their performance when they are conditioned on multiple frames. A more fine-grained per-experiment breakdown highlights that Veo3 outperforms Veo3-fast in almost all physical settings, but there is one notable exception, the holonomic pendulum (see Fig. 19 for detailsed per-experiment scores). Among the openly available models, WAN-2.1 with key-frame interpolation demonstrates superior performance, with an average total score of 0.55, close to proprietary models. However, since the last frame is provided, such prompting should be considered as an easier interpolation case vs. hard extrapolation for single and multi-frame conditioning. COSMOS-predict2 also shows good results, yielding the best performance in multi-frame conditioning and benefiting greatly from it both in reduced discard rates ($-9\%$) and overall scores ($0.43 \rightarrow 0.52$), and excelling in experiments like bouncing and rolling (see Fig. 19). PyramidalFlow, while occasionally excelling in specific scenarios (e.g., spring, $0.65$), exhibits inconsistent performance, with high variability and notably high discard rates. Other models show intermediate results.

The variability of discard rates across setups reflects the reliability of different models in generating physically plausible videos. Between models, average discard rates vary significantly, ranging from as low as $22.0\%$ for the Kling-turbo model to as high as $47\%$ (PyramidalFlow, single frame). The variance across experiments is also noticeable, as shown in Fig. 20, Fig. 21, and Fig. 22. Some experiments tend to be prone to only certain types of errors, e.g., stillness in sliding experiments. The analysis of the major reasons (see Fig. 6) behind high discard rates reveals the absence of motion (i.e. *stillness*) and the presence of duplicate objects, as well as, to a lesser extent, the disappearance of the object from the video. These persistent shortcomings in the models' abilities to produce consistent and realistic videos are well-known (Huang et al., 2024b).

Overall, all generated models exhibit substantial limitations compared to real-world performance (see Fig. 8), underscoring the significant gaps remaining in simulating physically accurate dynamics.

## 6 LIMITATIONS

**Morpheus** is restricted to Newtonian physics under controlled settings, which ensures reproducibility but limits coverage of broader physical processes. To simplify evaluation, we assume negligible air resistance and friction; however, this reduces realism and can penalize physically plausible generations when these assumptions are violated. Next, some invariances are inherently easier to satisfy, making purely observational evaluation less comprehensive. In the future, the benchmark could be extended to cover controllable VGMs, where it would be possible to measure how robust the invariance score is under perturbations. Finally, evaluation remains confined to short videos and simple scenes due to the scope of the vision foundational models' applicability and the static camera requirement.

## 7 CONCLUSION

Our study highlights a fundamental limitation in current video generation models: despite their impressive realism, they fail to consistently adhere to physical laws. To address this gap, we introduced Morpheus, a benchmark designed to assess the physical reasoning capabilities of these models. Through a curated dataset of real-world physics experiments and physics-informed evaluation

metrics, we demonstrate that even with advanced prompting techniques, existing models struggle to capture fundamental physical principles. In general, all models perform poorly, with significant violations of physical principles, though multi-frame prompting provides some improvement. This underscores the need for future research in integrating physical constraints into generative models.

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

# APPENDIX

## A  DATASET

Overall, we conducted a set of 9 core physical experiments, highlighting different physical principles, including:

1. Falling: Objects dropped from rest until they make impact with the surface, used to test uniform gravitational acceleration and energy conservation.

2. Projectile motion: A ball launched at various initial velocities and angles, testing the preservation of momentum and energy, as well as the uniformity of gravity.

3. Rolling: A metal can rolling from a slope, with energy conservation.

4. Sliding: A book sliding from a slope, with energy conservation.

5. Holonomic pendulum: A ball affixed to a rigid rod, with periodic motion and energy conservation.

6. Double pendulum: A more complex system with a pendulum attached to another pendulum, illustrating chaotic behavior and conservation laws in nonlinear dynamics.

7. Bouncing: A ball observed from the moment it first impacts the surface until it rebounds and impacts again, testing gravitational acceleration and energy conservation in a more challenging setting.

8. Collision: Two metal balls with the same or different masses collide with each other. One of them is initially stationary and the other one collides with it.

9. Spring: A weight hanging below a vertical spring, perform up and down simple harmonic vibration with energy conservation.

For each system, we recorded multiple times the type of experiment trying to have homogenous videos, while after a few iterations, we varied the initial conditions or configuration parameters. Table 1 below summarizes the number of recordings and configurations for each experiment.

| Experiment | Videos | Factors of Variation | Configuration Initial Condition Description |
|---|---|---|---|
| Falling | 20 | 2 | Object type and height from which the object was released. |
| Projectile | 15 | 3 | Angle of launch, slingback extension levels, launched ball color. |
| Bouncing ball | 12 | 1 | Heights from which the ball was released before bouncing. |
| Holonomic Pendulum | 22 | 1 | Initial angle from the vertical (zero-degree resting position). |
| Double Pendulum | 10 | 1 | Initial height of the second (top) pendulum bob. |
| Rolling | 15 | 2 | Incline angle of the ramp from which the object was released and object type. |
| Sliding | 10 | 1 | Incline angle of the ramp (slope) from which the object was released. |
| Collision | 12 | 3 | Masses of the colliding objects and their initial velocities before impact. |
| Spring | 7 | 1 | Magnitude of the initial force/impulse used to displace the mass from rest. |

Table 1: Summary of the experimental dataset from real-world recorded videos.

For each experiment, we provide representative frames: falling objects in Figure 9; bouncing, projectile motion, holonomic pendulum, and double pendulum in Figure 10; rolling in Figure 11; and sliding, collision, and spring in Figure 12.

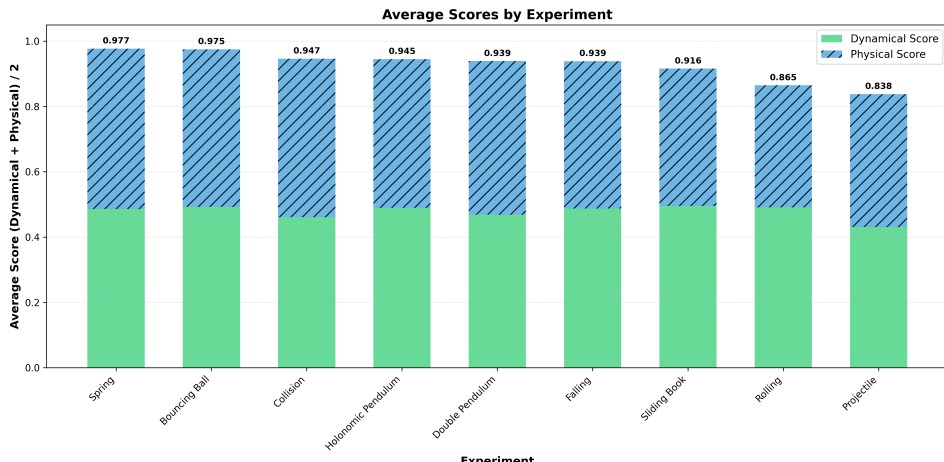

Figure 8: Morpheus scores for each experiment on real-world videos.

**Falling** For the falling experiment, we started with a standard table tennis orange ball as the simplest object. A mechanical actuator [1] was used to hold the ball at a certain height (initial position) and as a release mechanism to control the moment the ball fell free, before making contact with the surface below. Different height levels from the surface were used as initial positions, resulting in trajectories with different lengths (smaller or larger). In addition to the ball, we conducted extra experiments using different everyday objects, namely a plastic whiteboard marker, an adhesive tape roll, and an apple. Unlike actuator controlled experiments, these objects were released directly from the hand of a person at varying initial heights. This setup introduced additional variability in the conditions and the orientation of the object.

**Bouncing ball** The bouncing ball experiment begins immediately after the falling ball makes impact with the surface. It focuses on observing the ball during its bounce, capturing its trajectory as it rebounds upwards after contact with the surface.

**Projectile** For this experiment, a custom 3D printed projectile was built, along with three different balls of the same plastic material but of different colors. The projectile works with string rubber bands following the same principle of a slingback. During our recordings, we varied three different parameters. The angle of the launch for the ball, the force with which the ball was launched into the air, and the color of the ball.

**Holonomic pendulum** For this setting, a rigid metal structure consisting of a pole, perpendicular to the ground, on which a solid metal stick was mounted. The joint holding the stick was adjusted to allow for a normal friction coefficient, resulting in an intuitive retrogressive back-and-forth movement simulating a typical pendulum oscillatory trajectory. At the end of the metal stick a small table tennis ball was attached, as the SAM2 predictor can confidently track the center of the ball aligning with the central axis at the end of the stick. Using the zero angle as the resting position, we varied the angle at which the pendulum was released resulting in distinct retrogressive trajectories. As in the falling ball experiment the same release mechanism model was employed to manipulate the moment the pendulum was let freely to swing.

**Double pendulum** A custom structure consisting of a wooden base, a metal pole mounted on the top of the base, and a joint mounted at a degrees angle to the center axis of the pole, to keep the longer bob of the pendulum in place. These structures ensure that each 3D printed plastic bobs of the pendulum can rotate freely with normal friction resulting in the typical chaotic motion double pendulum are known for. A double pendulum consists of two bobs attached end-to-end. Each pendulum has its angle relative to the vertical. The same release mechanism as in previous

---

[1] *Motor Model: T825, Motor serial number: 00362129*

experiments is utilized to define the starting position of each pendulum link. This starting position can be described as the angle each bob makes with the vertical when it is still stationary.

**Rolling**  For this setting, we examined objects that roll down an inclined surface. We used three different objects: a full can, which was sealed, an empty can, and an orange. The slope of the surface was adjustable, allowing us to vary the steepness of the slope. The objects were placed by hand at the top of the slope before being released. Due to different mass distributions and shapes, we had different rolling behaviors across these three objects.

**Sliding**  For this experiment, we investigated the sliding motion on an inclined surface using a flat book. We varied the slope of the surface between trials. The book was placed by hand at the top of the slope before being released. Depending on the angle of the slope, the trajectories exhibited smooth sliding motion.

**Collision**  In the collision experiment, we studied collisions between objects of different sizes. Three setups were recoded: A large object collides with a smaller one, two objects of equal size collide with each other, and a small object collides with a larger one. For all cases, the initial velocity of the moving object (leftmost object with the right one at rest) was introduced with a gentle push at random speeds. These settings produced a diverse outcome for the aforementioned experiment, depending on the relative mass of the objects and the initial velocity.

**Spring**  In this spring experiment, we analyzed the oscillatory motion of a cylindrical metal weight suspended from a vertical spring. The object was initially displaced from its rest position by hand with a small (but random) force and then released. The amplitude of oscillation was defined by this displacement and the restoring force led the object to move in a vertical periodic movement until equilibrium. Initial force was the main reason for the motion, with no actuator being employed, and gradually over time decayed due to damping effects.

## B    AUGMENTATION WITH COSMOS-TRANSFER1

We augment a subset of experiments with style transfer to diversify object appearances while preserving underlying physical dynamics, creating initializations more representative of typical VGM training data. Specifically, we apply COSMOS-Transfer to five experiment types—Falling ball, Projectile, Holonomic pendulum, Rolling, and Sliding. For each experiment type, we select representative videos and generate style-transferred variants that alter object semantics and appearance while retaining motion consistency. This yields three transferred variants per original video (two for Holonomic pendulum), providing diverse yet physically plausible conditioning scenarios. Per experiment type, this produces $3 \times 3 = 9$ augmented conditioning scenarios for most experiments, and $3 \times 2 = 6$ for Holonomic pendulum.

The transfer process uses per-frame object masks from SAM2 Ravi et al. (2024), reusing the same masks from our trajectory-extraction pipeline to isolate only the object under study. We provide concise, object-focused prompts describing the desired replacement. All transfers undergo manual screening for visual and physical plausibility. When artifacts such as camera motion, temporal drift, hallucinations, shape misalignment, or mask leakage are observed, we enable Canny-edge guidance in COSMOS-Transfer with an edge-conditioning weight of 0.5, with the object mask now accounting for the other 0.5 conditioning weight. This additional constraint preserves scene structure while still allowing complete style transformation.

Figure 13 shows representative examples comparing the first frames of original videos with their style-transferred variants. These augmentations effectively multiply our conditioning scenarios: each original video contributes multiple physically consistent variants, expanding the diversity of initial conditions for VGM evaluation.

## C    VELOCITY AND ACCELERATION ESTIMATION

We estimate objects' velocity and acceleration from the extracted trajectory using multiple stages.

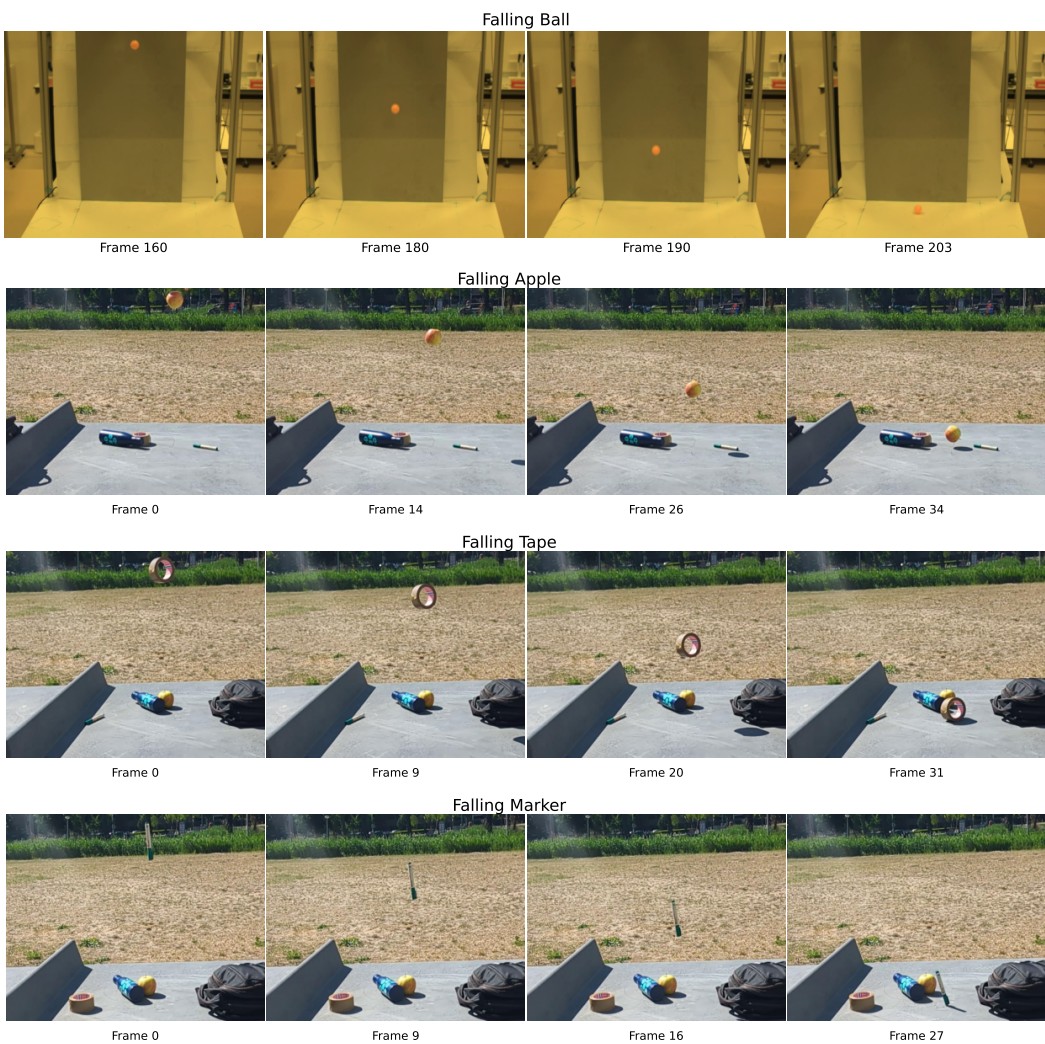

Figure 9: Representative frames from the falling experiments in the **Morpheus** benchmark: ball, apple, tape, and marker.

We use the central difference method for most points in the time series. This method computes velocity by considering both forward and backward positions, reducing single-sided differentiation errors.

$$v_i = \frac{x_{i+1} - x_{i-1}}{t_{i+1} - t_{i-1}}, \quad 1 \le i \le N - 2 \tag{1}$$

Since the central difference is not applicable at endpoints, we use one-sided differences. Forward difference (starting point):

$$v_0 = \frac{x_1 - x_0}{t_1 - t_0}$$

Backward difference (ending point):

$$v_N = \frac{x_N - x_{N-1}}{t_N - t_{N-1}}$$

To enhance precision, we perform linear regression within a sliding window.

$$x(t) = vt + b \tag{2}$$

The velocity (slope) is solved using the least squares method with window size w:

$$\begin{bmatrix} v \\ b \end{bmatrix} = \left(A^T A\right)^{-1} A^T x \tag{3}$$

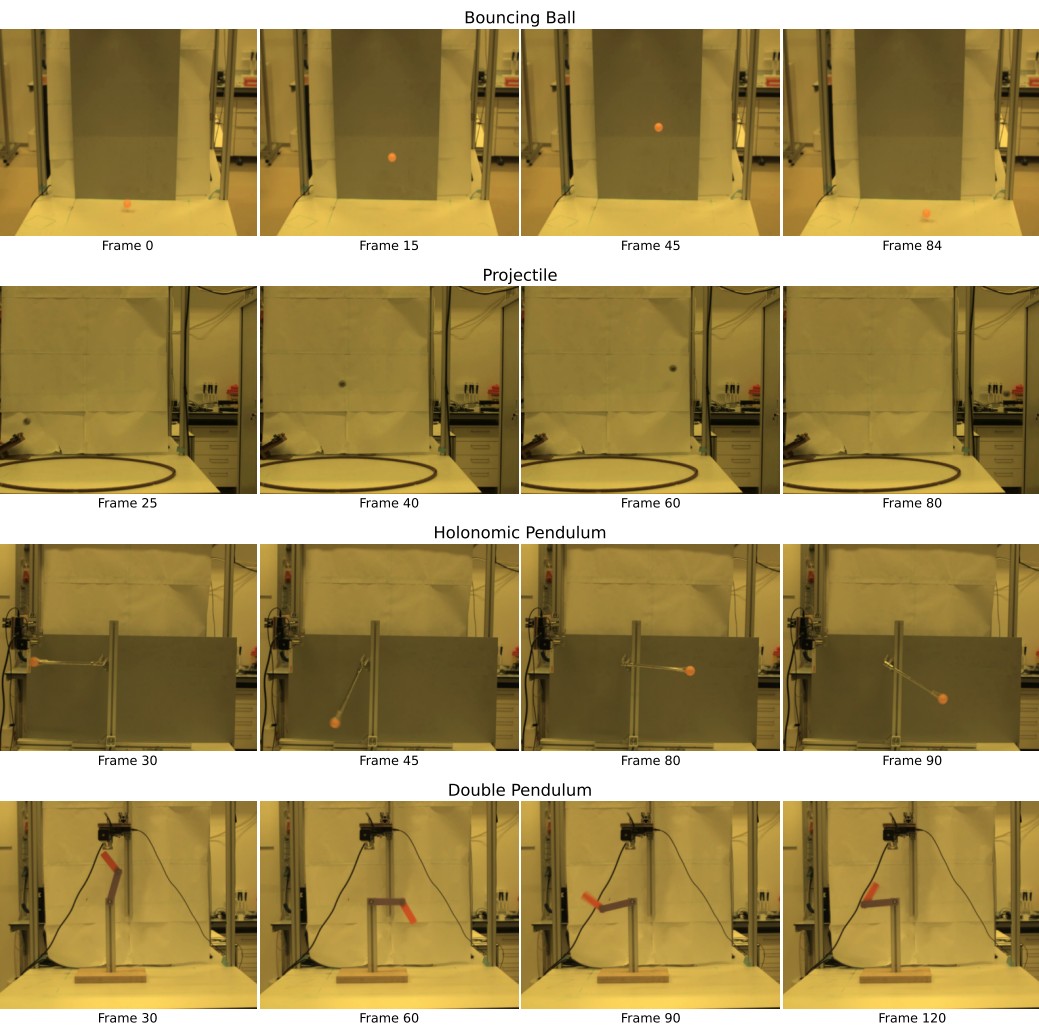

Figure 10: Representative frames from four experiments in the **Morpheus** benchmark: bouncing ball, projectile motion, holonomic pendulum, and double pendulum.

where matrix A contains time information.

$$A = \begin{bmatrix} t_1 & 1 \\ t_2 & 1 \\ \vdots & \vdots \\ t_w & 1 \end{bmatrix} \tag{4}$$

We combine linear regression and central difference results using weighted averages.

$$v_{\text{final}} = \alpha v_{\text{regression}} + (1 - \alpha)v_{\text{central}} \tag{5}$$

Here $\alpha = 0.7$, indicating greater confidence in the regression method. Finally, we apply Savitzky-Golay filtering for smoothing (Luo et al., 2005). This step effectively removes high-frequency noise from velocity calculations.

$$v_{\text{smoothed}} = \text{SG}(v_{\text{final}}, \text{window}, 3) \tag{6}$$

The entire calculation process can be summarized as:

$$v(t) = \text{SG}\left(\alpha v_{\text{regression}}(t) + (1 - \alpha)v_{\text{central}}(t), w, 3\right) \tag{7}$$

where w is the window size (odd number for symmetry); $\alpha = 0.7$ is the weighting coefficient; SG represents Savitzky-Golay filter of order 3; Regression window range: $[t - w/2, t + w/2]$.

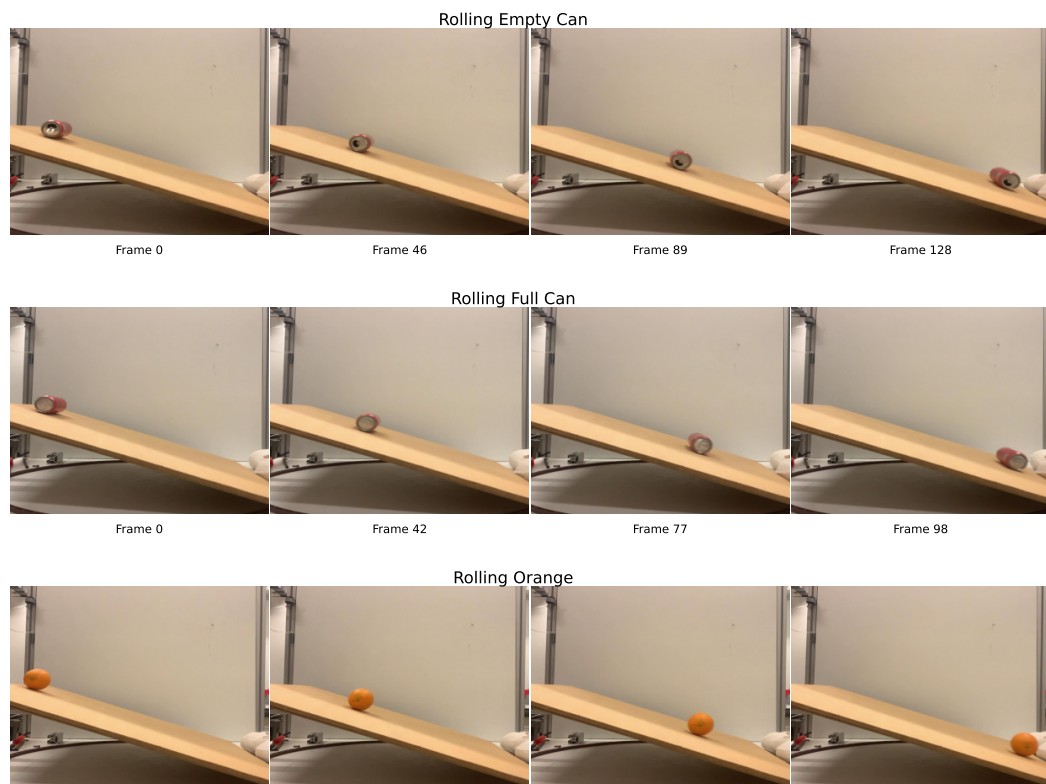

Figure 11: Representative frames of the rolling experiments in the **Morpheus** benchmark: an empty can, a full can, and an orange, each released by hand on an inclined surface of varying slope.

For the acceleration, we first calculate the acceleration using the central difference. For $1 \leq i \leq N-2$:

$$a_i = \frac{v_{i+1} - v_{i-1}}{t_{i+1} - t_{i-1}} \qquad (8)$$

Dealing with the endpoints using the same metric as velocities, we get the final acceleration for the entire trajectory.

$$a_0 = \frac{a_1 - a_0}{t_1 - t_0}$$

$$a_N = \frac{v_N - v_{N-1}}{t_N - t_{N-1}}$$

## D  EVALUATION METRICS

| Benchmark | Real-world ground-truth | Quantitative evaluation | Initial condition grounding | Physical laws evaluation |
|---|---|---|---|---|
| VideoCon-Physics | ✓ | ✗ | ✗ (only text) | ✗ |
| VAMP | ✓ | ✓ | ✗ (only text) | ✗ |
| PhyGenBench | ✓ | ✗ | ✗ (only text) | ✗ |
| Kang et al. (2024) | ✗ | ✓ | ✓ (1 or 3 frames) | ✗ |
| COSMOS | ✗ | ✓ | ✓ (image + video) | ✗ |
| Physics-IQ | ✓ | ✓ | ✓ (image + video) | ✗ |
| **Morpheus** (ours) | ✓ | ✓ | ✓ (image + video) | ✓ |

Table 2: Comparison of physics-based video understanding benchmarks. Our benchmark is the first to use real physical laws for evaluation. Symbols: ✓= supported, ✗= not supported

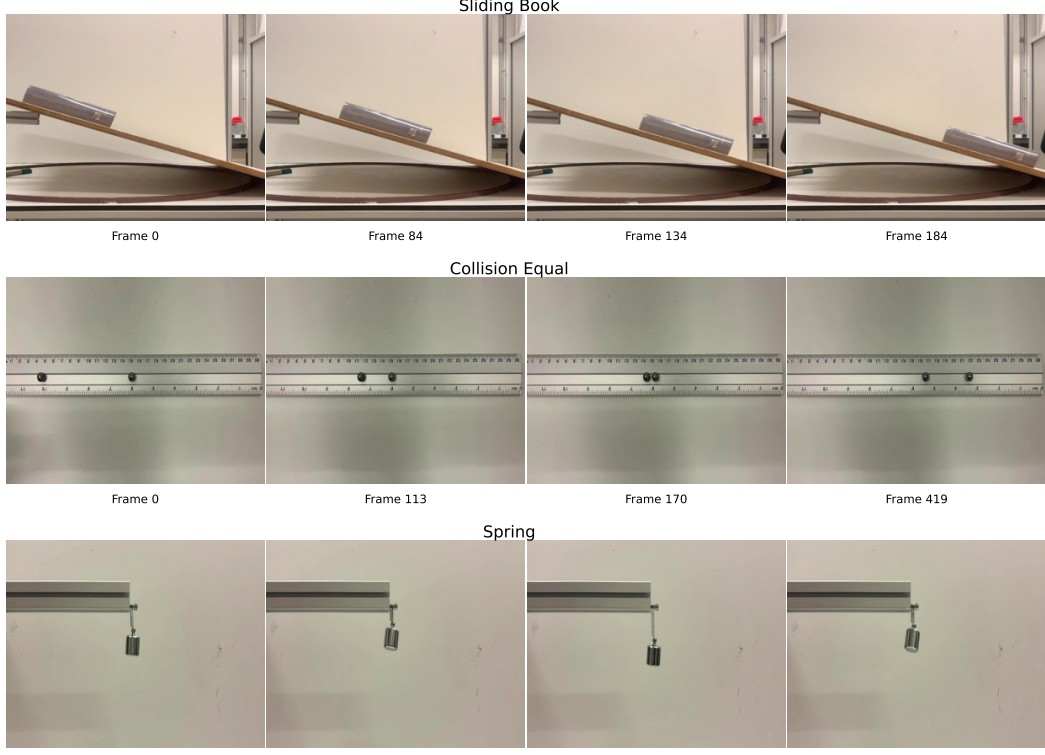

Figure 12: Representative frames from three experiments in the **Morpheus** benchmark: sliding, collision (equal-sized objects), and spring.

### D.1 DISCARD RATE

We generate $N_{total}$ videos for each type of experiment. Among these videos, we discard those that do not meet our quality standards, following a three-stage filtering out. First, we discard videos where object are disappearing from the videos the number of such videos is $N_{disappear}$. Second, we analyze the number of objects in each video and discard videos that do not maintain a consistent object count in the not discarded yet videos. For this purpose we employ DEVA tracking (Cheng et al., 2023) built on top of Grounded SAM (Ren et al., 2024) (with object names from the prompt as Grounding DINO (Liu et al., 2025) query) for consistent open-vocabulary prediction of 2D object masks. We denote the number of discarded videos in this step as $N_{duplicate}$. Specifically, we evaluate the proportion of frames containing multiple objects. Videos are filtered out if this proportion exceeds a predetermined threshold. Finally, we discard videos where the motion is too small to be meaningful in the not discarded yet videos, the number of such videos is $N_{stil}$. The overall *discard rate DR* is defined as

$$DR = \frac{N_{disappear} + N_{duplicate} + N_{still}}{N_{total}}.$$

### D.2 DEPTH CONSISTENCY EVALUATION

In all the studied experiments, the video camera is orthogonal to the object's motion and is fixed. This allows us to compute the Physical Invariance and Dynamical scores using only information extracted from 2D pixel space, available for generated videos. The results are presented in Fig. 15, showing that most of the models are reasonably consistent and thus object properties like energy conservation could be also studied using only 2D coordinates.

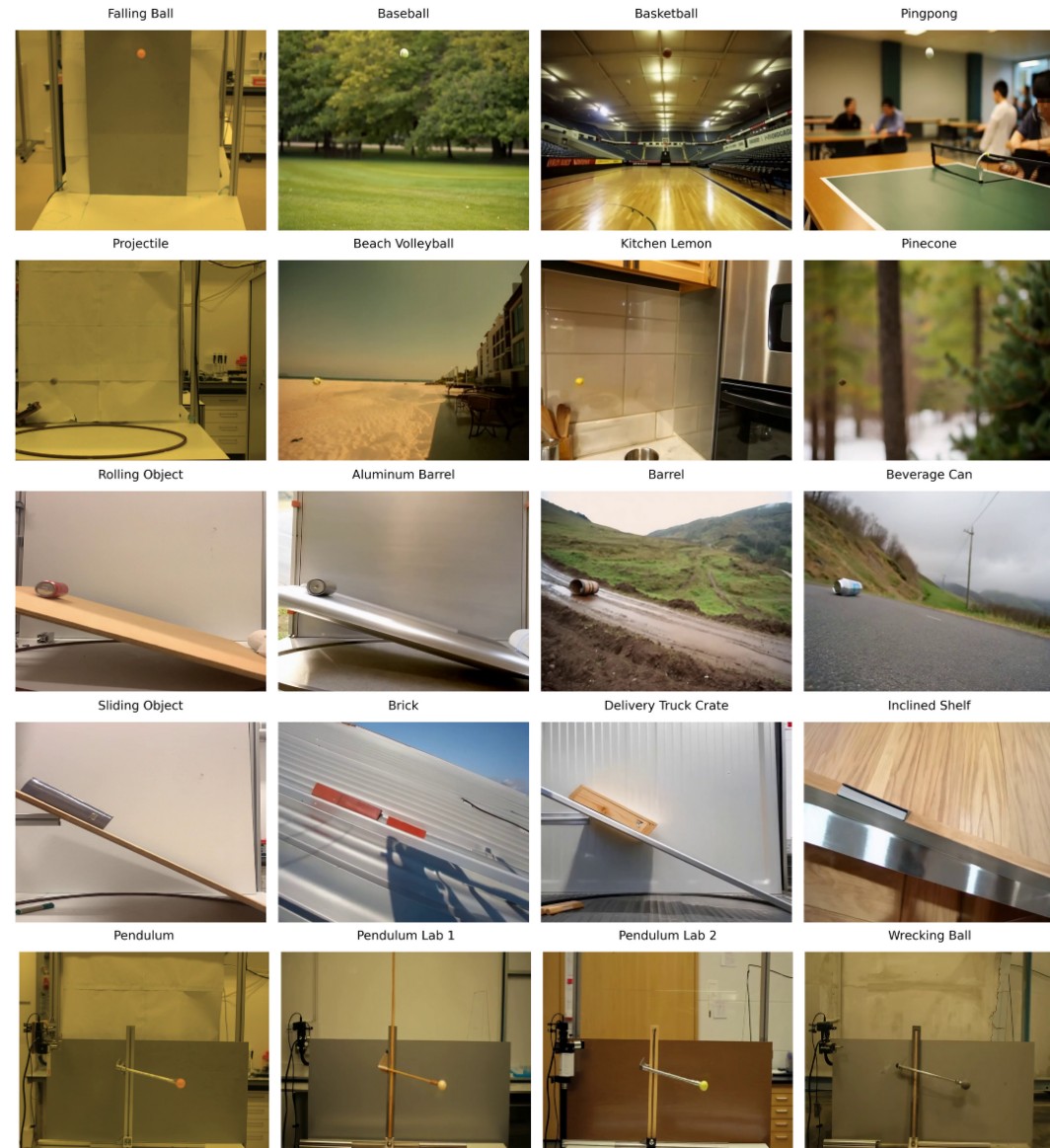

Figure 13: COSMOS-Transfer style augmentations for physics experiments. Each row displays the original experiment (left) with its style-transferred variants (right), showing how object appearance can be altered while preserving physical dynamics. The transfers provide diverse yet physically consistent initializations for VGM evaluation across five fundamental physics scenarios.

### D.3 DEPTH-CONSISTENCY VERIFICATION

To ensure that real-world and generated videos meet the static orthogonal camera assumption necessary to assess 2D trajectories as a projection of 3D motions, we incorporate a depth consistency check using Depth Anything V3. We calculate per-frame depth estimates and check that depth is roughly constant over time. This process verifies that the objects do not perform any unnatural motion along the camera's optical axis, which otherwise would invalidate physical measurements of velocity, acceleration or energy conservation computed only from the 2D pixel coordinates. Our metrics (Dynamical Score and Physical Invariance Score) depend on accurate 2D trajectories. For that reason the above depth consistency check is an important quality control step. This demonstrates that real-world recordings meet the static camera prerequisite. In addition, we also evaluate the same check with the newest Depth Anything V3, which enhances spatial sharpness and temporal

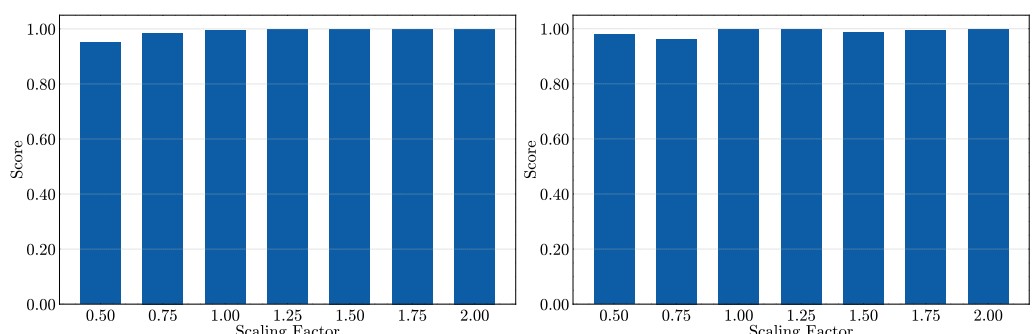

Figure 14: Scale invariance for Freefall (left) and Pendulum (right).

stability. DA2 and DA3 exhibit high depth stability across all physical experiments, with an average depth consistency scores of 98.3% and 96.8%, respectively. Each individual experiment exceeds 96% consistency for both models, which confirms that the objects are at constant distance from the camera throughout the motion. This confirms that all trajectories lie within a near planar imaging regime, justifying our use of 2D trajectories for physical evaluation. The strong agreement between DA2 and DA3 establishes that our conclusions are robust to this specific choice of depth estimators.

### D.4 SCALE INVARIANCE VERIFICATION

To confirm that our Dynamical evaluation pipeline is insensitive to precise camera calibration, we conducted a scaling experiment where each real world video trajectory was rescaled between a range of scaling factors (0.50, 0.75, 1.00, 1.25, 1.50, 1.75 and 2.00). Throughout this entire range the Dynamical score stays extremely close to 1, with minimal variations. The results for the pendulum, free fall and sliding experiments can be visualized in Figure 14. This clearly indicates that the PINN is capable of recovering the underlying physical dynamics when the absolute pixel changes within a reasonable range. In practice, this would mean that small differences in camera distance or object size do not affect the Dynamical Score. Therefore, no explicit camera calibration is needed for Morpheus and its evaluation is robust under natural scale variability in the real world setting.

### D.5 PHYSICALLY-INFORMED NEURAL NETWORKS

Unlike typical neural networks, which are normally trained only on data, prior knowledge about the physical system is integrated into PINNs. This prior knowledge of the physical system, often in governing physical laws such as Newtonian mechanics or energy conservation, is imposed during training. Given that the system modeled from the generated videos is known from the provided prompt, the training process incorporates these laws into the loss function. The total loss for a PINN is defined as:

$$L_{\text{total}} = L_{\text{data}} + \lambda L_{\text{physics}}, \tag{9}$$

where $L_{\text{data}}$ ensures that the network's output can match the observed data. At the same time, $L_{\text{physics}}$ is penalizing deviations from the governing physical equation and $\lambda$ is a hyperparameter balancing the contribution of the each loss component. For our own experimentation $\lambda$ has a value of 1. In this way, PINNs can bring both data and physical laws together during training while being consistent with the underline physical system. For a trajectory $T$, the data loss is defined as

$$L_{\text{data}} = \frac{1}{N} \sum_{i=1}^{N} \|\hat{T}_i - T_i\|^2, \tag{10}$$

, where $\hat{T}_i$ is the trajectory predicted by the network at the $i$-th timestep and $T_i$ is the corresponding ground truth trajectory at the same timestep. On the other hand, the physics loss is derived separately for each experiment, given the nature of the system's dynamics. The motion of a free-falling object follows:

$$\ddot{y} + g = 0, \tag{11}$$

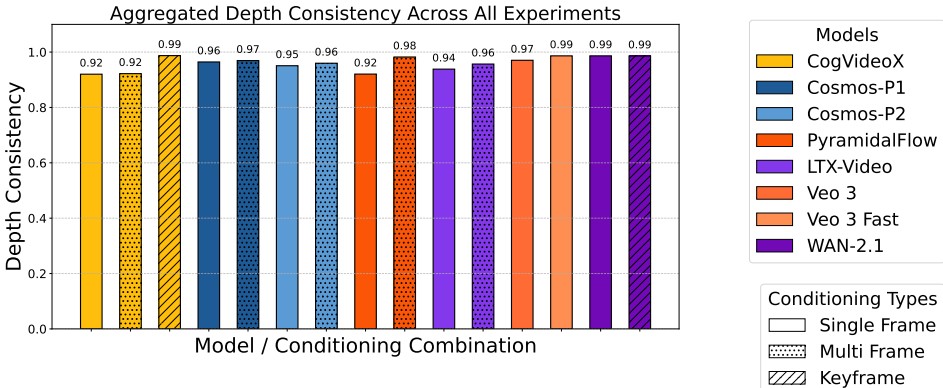

Figure 15: Average depth consistency for different video generation models across all studied experiments.

where $y$ is the vertical position, $\ddot{y}$ is the acceleration and $g$ is the gravitational constant. This means that for this phenomenon, the loss is defined as: The physics loss for free fall is defined as:

$$L_{\text{physics}} = \frac{1}{M} \sum_{j=1}^{M} \left\| \hat{\ddot{y}}_j + g \right\|^2, \tag{12}$$

where $\hat{\ddot{y}}_j$ is the predicted acceleration derived from the PINN at the $j$-th time step. The motion of a holonomic pendulum is governed by:

$$\ddot{\theta} + \frac{g}{l} \sin \theta = 0, \tag{13}$$

where $\theta$ is the angular displacement, $l$ is the pendulum length and $g$ is the gravitational constant.

The corresponding physics loss is:

$$L_{\text{physics}} = \frac{1}{M} \sum_{j=1}^{M} \left\| \hat{\ddot{\theta}}_j + \frac{g}{l} \sin(\hat{\theta}_j) \right\|^2, \tag{14}$$

where $\hat{\ddot{\theta}}_j$ and $\hat{\theta}_j$ are the network-predicted angular acceleration and displacement, respectively, at the $j$-th timestep. In the present work, we use the Dynamical score to evaluate how well the does the predicted trajectories align with the ground truth. The Dynamical score is derived from the Normalized Mean Squared Error (NMSE), which provides a relative measure of error by normalizing the Mean Squared Error (MSE) with the variance of the ground truth trajectory. Main motivation behind this choice, is to make the evaluation independent of scale. The NMSE is calculated as:

$$\text{NMSE} = \frac{\frac{1}{N} \sum_{i=1}^{N} (y_i - \hat{y}_i)^2}{\frac{1}{N} \sum_{i=1}^{N} (y_i - \bar{y})^2}, \tag{15}$$

where:

- $y_i$ is the true value at timestep $i$,
- $\hat{y}_i$ is the predicted value at timestep $i$,
- $\bar{y}$ is the mean of the ground truth values, defined as:

$$\bar{y} = \frac{1}{N} \sum_{i=1}^{N} y_i, \tag{16}$$

- $\frac{1}{N} \sum_{i=1}^{N} (y_i - \hat{y}_i)^2$ represents the MSE between the predicted and ground truth trajectories,
- $\frac{1}{N} \sum_{i=1}^{N} (y_i - \bar{y})^2 = \sigma^2$ represents the variance of the ground truth trajectory.

Table 3: Conserved quantities for each physical experiment in an ideal case.

| Experiment Name | Assumption | Conserved Quantities |
|---|---|---|
| falling ball | no air resistance | energy, acceleration (gravity), horiz. momentum |
| projectile | no air resistance | energy, acceleration (gravity), horiz. momentum |
| bouncing ball | no air resistance | energy, acceleration (gravity), horiz. momentum |
| holonomic pendulum | low resistance | energy, period, pendulum length |
| sliding | uniform surface forces | acceleration |
| rolling | uniform surface forces | acceleration |
| elastic collision | perfect elasticity | total energy, linear momentum |
| spring-mass system | ideal Hookean spring | period |
| double pendulum | low resistance | total energy, two pendulums length |

To ensure robustness, the predicted trajectory is compared against the interpolated ground truth values. Depending on the experiment, we address physical consistency by quantifying how well the learned solution adheres to the underlying physical equation. This is quantified using the physics loss, which penalizes deviations from the expected dynamics. For training, each PINN is optimized using the Adam optimizer with a learning rate of $10^{-3}$ for 200,000 iterations. The network used for all experiments, consists of two hidden layers of 20 neurons, with $\texttt{tanh}$ as activation functions. The final score is defined as $S_{dyn} = \min(1 - \text{NMSE}, 0)$.. Similarly to the Physical Invariance score, in cases when the original trajectory is discarded, the score is assigned to a minimal value equal to zero.

### D.6 Physical Invariances

**Falling Ball** For falling balls, energy must be conserved between consecutive bouncing points. Additionally, according to Newton's second law:

$$F = ma$$

In free fall, gravity is the sole force acting on the object, resulting in constant acceleration. Assuming that the gravitational field is uniform in space and time, we have $F = mg$, which means that $a = g$, so the acceleration should stay constant. Therefore, in this part, we introduce three quantitative metrics to assess trajectory physics: the Energy Conservation score (ES), which measures energy conservation within a specified time window, and the Acceleration Conservation score (AS), which evaluates the consistency of acceleration during this interval, and the Horizontal Momentum Conservation score (MS), which measures the conservation of momentum.

The Energy Conservation score is calculated as follows. Given the mass of the ball to be $m$, the $g$ a freefall acceleration constant, kinetic energy:

$$T = \frac{1}{2}m|\vec{v}|^2 = \frac{1}{2}m(v_x^2 + v_y^2)$$

and potential energy:

$$V = mgh$$

where $h = y$. Total energy is the sum of two:

$$E = T + V = \frac{1}{2}m(v_x^2 + v_y^2) + mgy$$

From this formula, assuming the mass of the ball is constant in time, we get:

$$\frac{E}{m} = \frac{1}{2}(v_x^2 + v_y^2) + gy = \text{const} \tag{17}$$

The calculation of the Acceleration Conservation score is self-evident:

$$a = \text{const} \tag{18}$$

The conservation of horizontal momentum arises from the fact that the only force acting on the ball is gravity, which is pointed downwards:

$$p_x = mV_x = \text{const}$$

and analogous to the energy, we deduce:

$$\frac{p_x}{m} = V_x = \text{const} \tag{19}$$

We provide some examples of estimated invariants in Fig. 26.

**Projectile** For projectile motion, we analyze the same physical invariants as in the falling ball experiment. Throughout the projectile's trajectory, neglecting the air resistance, energy, acceleration, and horizontal momentum should be conserved. The calculations for energy and acceleration follow the same methodology used in the falling ball analysis.

**Holonomic Pendulum** For the holonomic pendulum, let's first examine energy conservation. Energy in the ideal (frictionless) case:

$$H = T + V = \frac{p_\theta^2}{2mL^2} + mgL(1 - \cos\theta)$$

where $\theta$ is the angular displacement, $l$ is the pendulum length, $g$ is the gravitational acceleration, and $p_\theta = mL^2\dot{\theta}$ is the momentum.

In this case, the equation that we obtain is:

$$\ddot{\theta} + \frac{g}{l}\sin\theta = 0$$

Since our real-world pendulum experiments were conducted in a laboratory environment, friction causes energy attenuation over time. We quantify this energy loss by measuring both its range and rate of decline, establishing these as upper bounds for evaluating generated videos. To be specific, the holonomic pendulum with friction can be expressed as

$$\ddot{\theta} + \frac{b}{m}\dot{\theta} + \frac{g}{l}\sin\theta = 0 \tag{20}$$

where $b$ is the damping coefficient, $m$ is the bob mass, and $\frac{b}{m}\dot{\theta}$ represents the damping force term. The energy decay over time:

$$\frac{dE}{dt} = -b(\dot{\theta})^2 \tag{21}$$

In our experiments, we assume that the energy loss can be ignored for a short time period, meaning we can apply the Energy Conservation score.

The period of holonomic pendulum with friction with a small amplitude can be expressed as

$$T = 2\pi\sqrt{\frac{l}{g}}\sqrt{1 - \left(\frac{b}{2m\omega_0}\right)^2} \tag{22}$$

where $\omega_0 = \sqrt{\frac{g}{l}}$ is the natural angular frequency without damping. When the damping is small ($b \ll m\omega_0$), the period approaches that of an undamped pendulum $T_0 = 2\pi\sqrt{\frac{l}{g}}$. We observe this regime in our experiments and propose to use the Period Conservation score (PC).

For the holonomic pendulum, it is obvious that the pendulum length $l$ remains constant throughout the experiment, as the holonomic constraint of the system. Therefore, we also consider the pendulum length as a physical invariant.

## D.7 PHYSICAL SCORE SCALING

When we obtain the physical invariant value $C$, we calculate the relative standard deviation over time:

$$C_{\bar{\sigma}} = C_\sigma / C_{mean} \tag{23}$$

To ensure that the score is within the $[0, 1]$ range, we design the Physical score, derived from the invariant, as follows

$$S = \frac{1}{1 + \alpha * C_{\bar{\sigma}}} \tag{24}$$

Where $\alpha$ is a normalization factor. In the experiment, we set it to 1.0.

Two critical considerations emerge during the score calculation process. First, The time window must be carefully selected. For each trajectory, we partition it using a sliding time window and select the highest score among all segments as the trajectory's overall score. This approach addresses a key challenge in real-world experiments like bouncing balls, where fluctuations near bouncing points create large standard deviations and low scores. By using the highest score across all segments, we effectively capture the most stable portion of the trajectory.

In our experiments, we set the time window length equal to 25% of the total trajectory duration. Specifically, we use $t_{window} = L_{trajectory}/4$. In addition, proper scaling is essential: since the trajectory coordinates are recorded in pixel space rather than real-world 3D coordinates, a precise coordinate transformation to physical units is required. Notably, improper scaling can significantly impact the total energy calculations. Third, we need to be careful not to choose the range when the mean of the selected physical invariant is near zero. The absolute value of mean of the selected physical invariant should be equal to or greater than a threshold of 10 times of standard deviation:

$$C_{threshold} = 10 * C_{\sigma} \tag{25}$$

so it can be neglected that the influence of mean energy/acceleration is near zero. In the experiment, if $|C_{mean}| \geq C_{threshold}$, we calculate the Physical score as defined in Eq. 24. Otherwise, we use the following Eq. 26 that takes the absolute standard deviation rather than the relative standard deviation.

$$S = \frac{1}{1 + \alpha * C_{\sigma}} \tag{26}$$

This method has two key limitations. First, the scores are highly sensitive to the choice of time window size. Larger time windows tend to yield lower scores as they encompass more fluctuations in the trajectory. Thus, we kept time window constant between evaluating different model generations. Second, the method may fail to detect unphysical behavior in generated videos where objects remain stationary for long periods; however, this is addressed by discarding videos due to stillness.

## E    EXTENDED LIMITATIONS

The main focus of our benchmark is towards a set of Newtonian scenarios, which take place under a controlled environment and static camera conditions. While this enables clean invariants and can be reproducible, it limits a full scale physical coverage. Additionally, ambiguity is introduced (e.g. distortion of lens, unknown scale, etc.) because all of our estimates come from 2D pixel trajectories without any camera calibration. At the same time, we rely on assumptions such as negligible air resistance and friction. Unmodeled forces, such as air drag, friction and rotational kinetic energy could cause violations, which do not reflect generations that are wrong but rather evaluator mismatch. We perform trajectory extraction from generated videos using fixed first/last frame or short multi-frame conditioning that cannot specify initial conditions fully, such as the mass of an object or the coefficient of the spring. Moreover, a physically plausible generation can be penalized heavily when some of the assumptions are violated, leading to misleading results. Similarly, one single invariant can look really "good", but the generated video could violate multiple laws, making it physically implausible.

Errors in segmentation and tracking introduce noise. Additionally, we are only considering short clips, and the model might drift away from physical laws over a longer temporal horizon, which our current metrics would not be capable of capturing. Finally, it should be noted that while our scores remain proxies rather than direct measurements of underlying physical plausibility, that would require falsifiable and controllable VGM generation. Prior benchmarks that rely on judgments from humans or VLMs are prone to hallucinations and often miss slight inconsistencies. In contrast, our scores avoid these issues but still need to be interpreted jointly to give a much more reliable picture. With that in mind, Morpheus should be treated as a focused and reproducible stress test for core physical

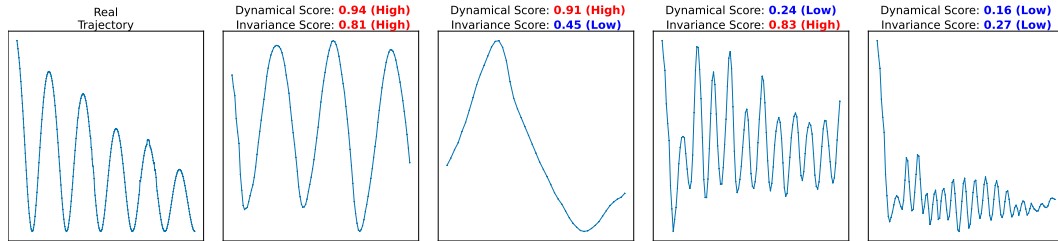

Figure 16: Real pendulum trajectory alongside four cases of generated videos, demonstrating how different combinations of dynamical and physical invariance scores appear in practice.

laws and not a benchmark that can test and evaluate all physical dynamics that might take place in a single video.

## F  VIDEO GENERATIVE MODELS DETAILS

At their core, latent video generative models often utilize a combination of a 3D Variational Autoencoder (VAE) (Kingma, 2013; Lin et al., 2024) to tokenize individual frames, a text encoder, like T5 (Raffel et al., 2020) to encode frames into latent. During training a noisy latent is produced by the forward diffusion process. This latent is then processed by a parametrized model, either a transformer model (Yang et al., 2023) or a U-Net (Ronneberger et al., 2015; Zhou et al., 2022; Bar-Tal et al., 2024) resulting in a patchified long sequence of visual tokens, in case of the former type of model.

Depending on the model architecture, different input modalities can be handled like text-to-video, image-to-video, text+image-to-video, and sometimes video continuation regimes facilitating both open-domain and controlled generation scenarios (Yang et al., 2024b; Kong et al., 2024; Agarwal et al., 2025; Brooks et al., 2024b). Although some state-of-the-art video generation models adopt an autoregressive framework, predicting frames sequentially based on prior outputs (Weng et al., 2024; Deng et al., 2024; Weissenborn et al., 2020), many others utilize non-autoregressive approaches to generate frames simultaneously (Yang et al., 2024b; Kong et al., 2024; Xing et al., 2025). In Table 4, we specify the parameters of particular models used in our benchmark, along with it's architectural design choices. As to faithfully get the best generation outcome we use the best hyperparameters reported for each model.

| Model [Params.] | Resolution | Number of Video Frames | Guidance Scale | Sampling Steps |
|---|---|---|---|---|
| CogVideoX [5B] | 960 x 768 | 84 | 6.0 | 50 |
| Cosmos-Predict 1 [14B] | 1280 × 704 | 121 | 7.0 | 35 |
| Cosmos-Predict 2 [14B] | 1280 × 704 | 93 | 7.0 | 35 |
| WAN2.1 [14B] | 1280 × 720 | 121 | 5.0 | 40 |
| LTX-Video [13B] | 960x736 | 81 | 3.0 | 50 |
| PyramidalFlow[12B] | 1280 x 768 | 121 | 4.0 | 10 |

Table 4: Details of video generation models adopted in our benchmark study, including their resolution, number of video frames, guidance scale, and sampling steps.

## G  PROMPTS FOR VIDEO GENERATION MODELS

For each experiment, we carefully designed a prompt that describes the physical setup and motion of the experiment being conducted. For example, in the falling ball experiment, the prompt specifies that the ball falls and makes contact with the table below. Similarly, in the projectile experiment, we describe how the ball is launched at a slight upward angle and follows a natural parabolic trajectory. We enhance these prompts using an internally provided upsampler or ChatGLM if no upsampler is provided to incorporate more detailed scene descriptions and contextual elements derived from the reference images. All prompts are shown in Table 5 and Table 6.

| Experiment Name | Base Prompt | Enhanced Prompt |
|---|---|---|
| **Falling Ball** | Orange ping-pong ball falling down and making impact with the table surface below. Fixed camera view, no camera movement. | A ping-pong ball is captured in mid-air, suspended above a laboratory table, poised to make contact with the surface below. The ball's descent is governed by the force of gravity, creating an arc that suggests a controlled experiment in progress. The backdrop is a stark, clinical room with a neutral palette, punctuated by the sterile lines of a metal frame and the functional design of a nearby cabinet. The lighting is subdued, casting a soft glow that highlights the ball's trajectory and the anticipation of impact. The table beneath the ball is marked with faint lines, perhaps indicating measurements or guidelines for the experiment. As the ball continues its downward journey, it will likely bounce off the table, adding a dynamic element to the scene and marking the conclusion of this controlled descent. Fixed camera view, no camera movement. |
| **Projectile** | A single, small 3D-printed ball, dark gray in color, is launched from a plastic, small-scale ramp with a slight upward angle. The ball follows a natural, smooth, arcing trajectory upward and then downward, continuing along that arc until it exits the right side of the video frame. The video should accurately simulate the ball's motion under standard earth gravity, showing a clear parabolic arc. The video should emphasize a smooth and realistic physics-based movement of the ball without any sudden changes in speed. The ball should not bounce or collide with any objects in the scene. Fixed camera view, no camera movement. | In a meticulously crafted scene, a solitary, dark gray 3D-printed ball, with its sleek, spherical form, is propelled from a plastic ramp that slopes gently upward. The ball, weighing a mere fraction of a kilogram, is captured in high-definition, showcasing every nuance of its motion. As it leaves the ramp's edge, the ball arcs gracefully into the air, its trajectory a perfect parabola that mirrors the laws of physics under standard earth gravity. The video's frame follows the ball's smooth ascent and descent, highlighting the ball's consistent speed and the absence of any sudden accelerations or decelerations. The scene remains unobstructed, ensuring that the ball's journey is uninterrupted by any external forces, save for the pull of gravity, resulting in a visually stunning and scientifically accurate demonstration of a parabolic motion. Fixed camera view, no camera movement. |
| **Rolling Can** | An empty soda can rolls down a wooden surface. Experiment carried out in a laboratory controlled environment. Fixed camera view, no camera movement. | An empty aluminum red soda can rolls steadily down an inclined wooden board in a controlled laboratory environment. This motion should be depicted with physical realism, accurately simulating the key dynamics: the can accelerates under gravity, with its combined rotational and translational movement appearing authentic and governed by the frictional interaction with the wooden surface, ensuring a physically plausible rolling movement. Fixed camera view, no camera movement. |
| **Sliding Book** | A book slides down a wooden surface. Experiment carried out in a laboratory controlled environment. Fixed camera view, no camera movement. | A book slides steadily down an inclined wooden board in a controlled laboratory environment. This motion should be depicted with physical realism, accurately simulating the key dynamics: the book accelerates due to the component of gravity acting along the incline, opposed by kinetic friction from the wooden surface. Its movement should be purely translational, maintaining consistent contact with the board and without any significant rotation or tumbling, ensuring a physically plausible descent. Fixed camera view, no camera movement. |
| **Holonomic Pendulum** | A single pendulum moving retrogressively back and forth. At the bottom of a pendulum, there is a ball attached to it. The pendulum is holonomic. Fixed camera view, no camera movement. | A pendulum with a spherical ball attached swings back and forth in a controlled manner, its motion captured in a moment of retrograde swing. The pendulum's arm, likely made of metal, extends horizontally from a stand, connected to a pivot point that allows for rotational movement. The ball, positioned at the lower end of the pendulum, appears to be in motion, indicating the pendulum's swing. The environment suggests a laboratory or testing setting, with a backdrop of technical apparatus and equipment, and the lighting is artificial, casting a uniform glow over the scene. The pendulum's movement, while currently in a retrogressive swing, could potentially change direction, continuing its oscillatory motion. Fixed camera view, no camera movement. |

Table 5: Base and enhanced textual prompts used for video generation experiments: Falling Ball, Projectile, Rolling Can, Sliding Book, Holonomic Pendulum. Enhanced prompts are generated using corresponding upsampler VLMs such as ChatGLM (Zeng et al., 2024) and incorporate more detailed scene descriptions and contextual elements derived from the reference images. Slight modifications to these prompts were made in case of different variants such as object type.

| Experiment Name | Base Prompt | Enhanced Prompt |
|---|---|---|
| **Double Pendulum** | Double pendulum, consisting of a purple and an orange segment. Each segment moves independently. Fixed camera view, no camera movement. | In a meticulously arranged laboratory setting, a double pendulum setup swings gracefully, each pendulum segment adhering to the immutable laws of physics. The upper pendulum, a sleek purple rod, contrasts strikingly with the lower orange rod, both suspended from a sturdy, metallic frame. The room is bathed in soft, ambient light, casting subtle shadows that accentuate the pendulums' arcs. The scene captures the intricate dance of the pendulums, their movements a mesmerizing testament to the natural order, with each swing a silent symphony of motion and balance. Fixed camera view, no camera movement. |
| **Bouncing Ball** | A single orange ping pong ball bounces vertically as a result of making impact with the table after being in free fall. The ball starts in the center of the frame, and moves upwards. Fixed camera view, no camera movement. | A solitary orange ping pong ball, with its vibrant hue standing out against the stark white of the table, plummets from the center of the frame. As the ball bounces upwards, it arcs gracefully, the trajectory a perfect parabola. The frame remains centered, emphasizing the ball's solitary dance of motion and the physics of its rebound. Fixed camera view, no camera movement. |
| **Collision** | Generate a realistic video of two metallic spheres colliding. In the first frames the leftmost sphere in the video is moving towards the static one. At the moment of contact the ball in motion transfer its kinetic energy to the ball at rest. The two spheres have identical physical properties, namely material, shapes, masses. The momentum should be conserved. Fixed camera view, no camera movement. | In a high-definition, slow-motion sequence, two identical metallic spheres, each polished to a mirror-like finish, are meticulously positioned on a frictionless surface. The left sphere, propelled by an unseen force, hurtles towards the stationary sphere, its trajectory a perfect parabola. As the oncoming sphere approaches, the static sphere begins to subtly vibrate, a prelude to the impending collision. The moment of contact is captured in stunning detail, revealing the transfer of kinetic energy as the moving sphere's momentum is transferred to the stationary one. The spheres, crafted from the same dense material, maintain their spherical shapes and masses, ensuring the conservation of momentum throughout the impact. The scene is illuminated by a single, soft light source, casting long shadows and emphasizing the physics of the collision. Fixed camera view, no camera movement. |
| **Spring** | A single metallic slotted mass attached on a spring moving periodically up and down. Fixed camera view, no camera movement. | In a meticulously crafted scene, a solitary metallic slotted mass, polished to a gleaming silver, is ingeniously attached to a tensioned spring. The mass, weighing several pounds, oscillates with a fluid grace, its movement initiated by the subtle release of the spring. The camera captures the mass as it arcs upwards, the tension in the spring visible, before it gently descends with a rhythmic sway, the sound of its metallic slats clinking softly in the background. The scene is set against a stark white backdrop, emphasizing the stark contrast between the mass and the spring, and the smooth, periodic motion of the mechanical dance. Fixed camera view, no camera movement. |

Table 6: Base and enhanced textual prompts used for video generation experiments: Double Pendulum, Bouncing Ball, Collision, Spring. Enhanced prompts are generated using corresponding upsampler VLMs such as ChatGLM (Zeng et al., 2024) and incorporate more detailed scene descriptions and contextual elements derived from the reference images. Slight modifications to these prompts were made in case of different variants such as object type.

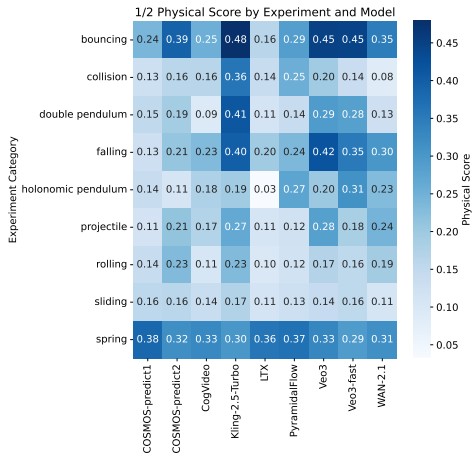

Figure 17: The distribution of the average physical invariance score across models and experiments.

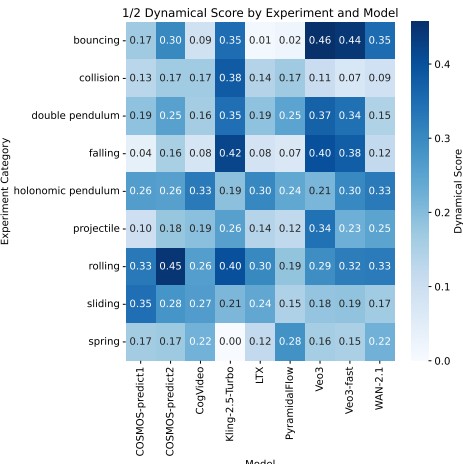

Figure 18: The distribution of the average dynamical score across models and experiments.

## H  ADDITIONAL ANALYSIS

In Figure 26, we present an additional visualization of the energy and acceleration conservation. In Figure 16, we visualize the difference between the object trajectories of real-world and generated videos. In Figure 26(a), we present the total, kinetic, and potential energy over time. As expected, the total energy dissipates with every new bounce while remaining nearly constant between bounces.

## I  USAGE OF LARGE LANGUAGE MODELS

Large language models were used solely for grammar correction and language polishing of the manuscript. They had no involvement in the research design, experimentation, analysis, or generation of results; all technical contributions are the work of the authors.

**Ethics statement.**  This work involves the collection of benchmark datasets and the evaluation of publicly available models, all used in compliance with their original licenses. No new human data or personal annotations were gathered. Our study focuses exclusively on benchmarking and reproducibility, without any collection or processing of user data.

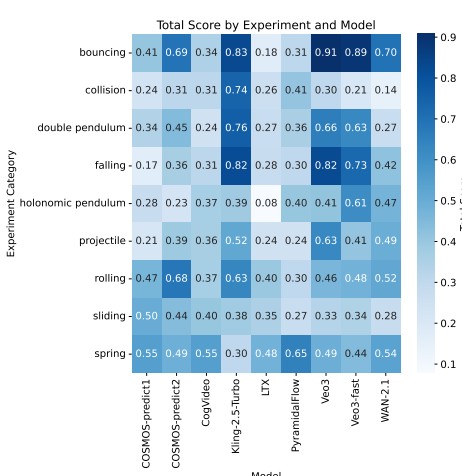

Figure 19: The distribution of the average total score across models and experiments.

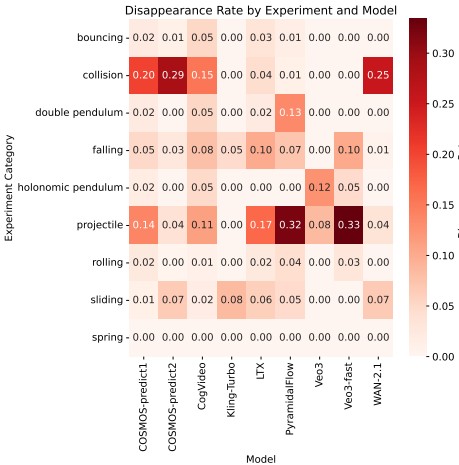

Figure 20: The distribution of the average discard rate due to the **objects' disappearance** across models and experiments.

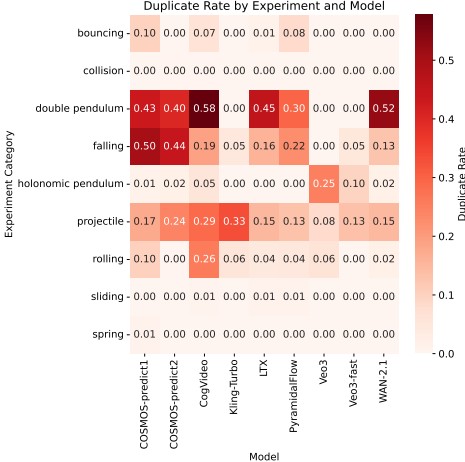

Figure 21: The distribution of the average discard rate due to the **objects' duplicates** across models and experiments.

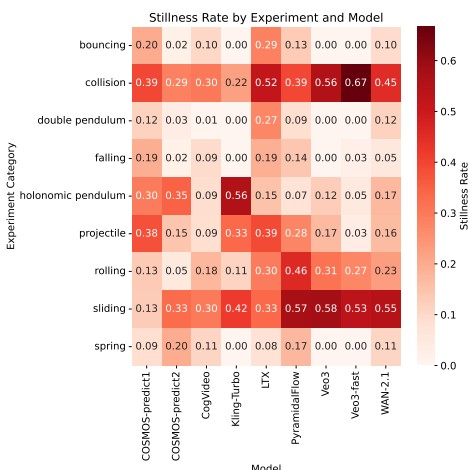

Figure 22: The distribution of the average discard rate due to **objects' stillness** across models and experiments.

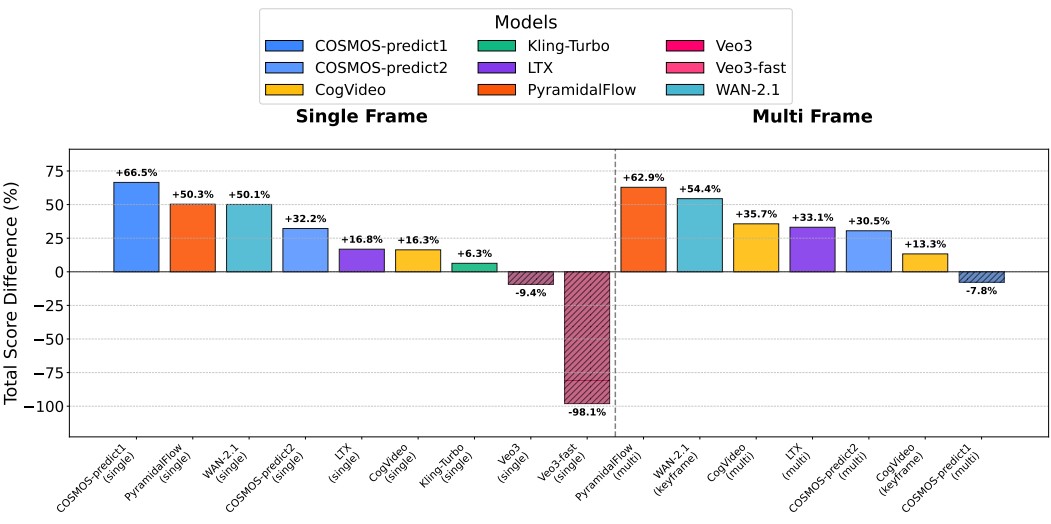

Figure 23: The relative change in scores evaluating on original data vs. on the augmented data. The augmented examples seem to be much harder.

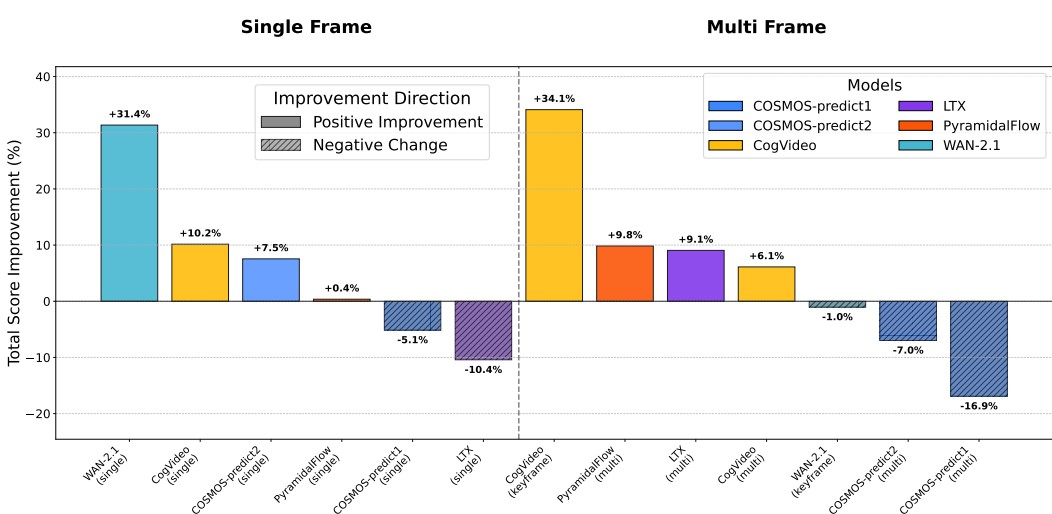

Figure 24: The relative change in scores for using an enhanced prompt vs the plain one.

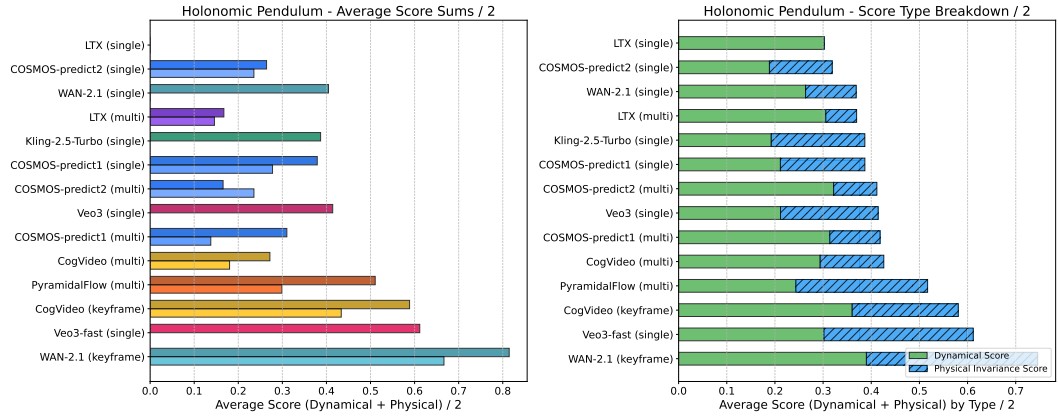

Figure 25: The scores for the holonomic pendulum experiment. On the left, darker colors denote *enhanced* textual prompt.

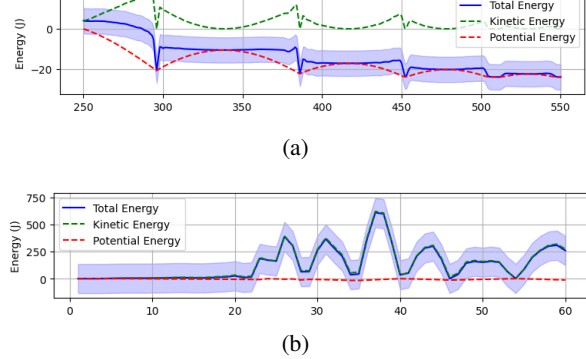

Figure 26: Energy analysis of real-world and generated falling + bouncing ball videos: (a) Real-world video energy conservation (b) CogVideoX plain single frame generated video energy conservation

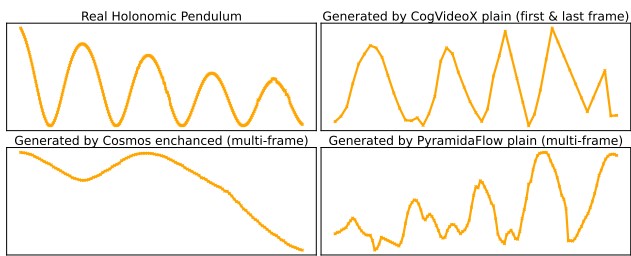

Figure 27: Real (top left) and generated trajectories for the holonomic pendulum.

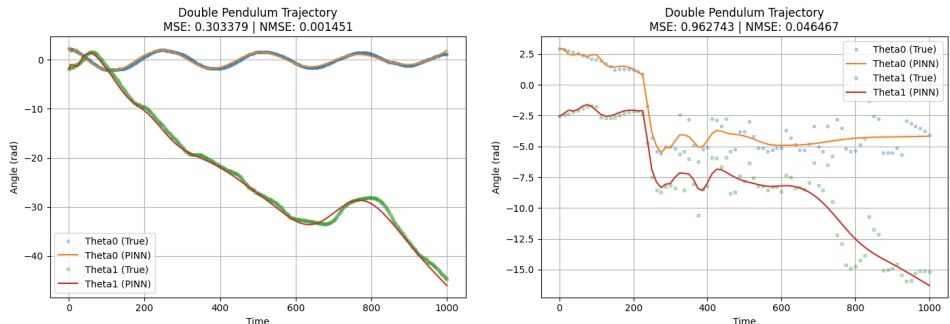

Figure 28: Real (left) and generated trajectory (right) for the double pendulum and corresponding fitting curve with PINN. While NMSE for generated trajectory is small 0.05, it is still 50 times worse that PINN with the same parameters fitted to real-world trajectory.

**Reproducability statement.** For reproducibility, we release the full code on GitHub and all benchmark data on Hugging Face.

