# OpenReview forum: "Benchmarking Physical Reasoning of Video Generative Models with Real Physical Experiments"
_ICLR.cc/2026/Conference — Submitted to ICLR 2026_

### Official Review · Reviewer_LyJG · 2025-10-31

**Soundness:** 3
**Presentation:** 2
**Contribution:** 2
**Rating:** 6
**Confidence:** 4

**Summary:**

This paper presents Morpheus, a benchmark for evaluating the physical reasoning ability of video generative models (VGMs). Using 130 real-world physics experiments and physics-informed neural networks (PINNs), it measures whether generated videos obey Newtonian laws and conserve physical invariants like energy and momentum. The benchmark computes two metrics—Dynamical Score and Physical Invariance Score—to quantitatively assess physical plausibility. Experiments on major VGMs (e.g., COSMOS, Wan2.1, LTX, PyramidalFlow) show that, despite realistic appearances, current models fail to respect fundamental physical principles, highlighting the need for physically grounded video generation.

**Strengths:**

1. The paper introduces the first benchmark grounded in real physical experiments, offering a rigorous and reproducible setup beyond simulation or human judgment.

2. The use of PINN-based evaluation metrics to measure both dynamical consistency and physical invariants is technically sound and interpretable.

3. The analysis is comprehensive, covering multiple models, prompt settings (text, single/multi-frame), and experiments, providing broad insights into current limitations of VGMs.

**Weaknesses:**

1. The main limitation lies in scope and generality. The benchmark only covers basic Newtonian systems under controlled lab conditions, excluding non-linear, stochastic, or complex real-world settings (e.g., fluid, thermodynamic, or deformable-body dynamics). This makes it less reflective of broader physical reasoning in the wild.

2. The metrics rely heavily on accurate object tracking via SAM-2 and Depth Anything. Noise, occlusions, or tracking drift could propagate into PINN fitting, potentially biasing the scores. While the paper mentions smoothing and filtering, there is no sensitivity analysis showing metric robustness to visual noise or tracking failure.

3. While Morpheus claims objectivity, the design of invariants and ODE priors is manually specified. This limits scalability—each new physical process requires manual equation definition. Integrating data-driven symbolic discovery or learned physics priors could make the benchmark more generalizable.

4. The prompt engineering process (using upsamplers and ChatGLM-based expansions) introduces confounding factors. It is unclear whether performance differences arise from physical reasoning capability or prompt quality. The fairness of cross-model comparisons (especially between keyframe interpolation vs. open-ended generation) also remains questionable.

**Questions:**

1. How sensitive are the Morpheus scores to tracking or depth estimation errors in the trajectory extraction pipeline?

2. Are the PINN hyperparameters (e.g., λ for physics loss) fixed across all experiments, and how do they affect comparability?

3. Does Morpheus evaluate 3D consistency (e.g., parallax, occlusion) or only 2D motion projections?

4. Can this framework be extended to evaluate non-Newtonian or fluid-based systems?

**Details Of Ethics Concerns:**

Nan

---

> ### Author Response · Authors · 2025-11-28
>
> We sincerely thank Reviewer LyJG for their detailed evaluation. We are encouraged that the reviewer recognized Morpheus as the "first benchmark grounded in real physical experiments" and "rigorous and reproducible setup" and the "technically sound" use of PINN-based metrics. Below, we address the reviewer's specific concerns on scope, methodology, and fairness.
>
> **W1: Scope and Generalization**
>
> **Reviewer's Comment:** *The benchmark only covers basic Newtonian systems... excluding non-linear, stochastic, or complex real-world settings...*
>
> **Our Response:**
> The scope of physical phenomena was also raised by other reviewers and we clarify here that our focus on Newtonian dynamics is a deliberate scientific choice. To keep the response consistent across reviewers, we summarize our rationale and how the framework generalizes beyond its current score.
>
> We acknowledge that we focused on Newtonian mechanics. As detailed in our general comment, even proprietary SOTA models (Kling-2.5-Turbo, Veo3 and Veo3-fast) **fail to consistently solve such *simple* scenarios**, struggling with basic conservation laws like energy and momentum. Until models can master these ABCs of physics, evaluating them on *complicated physics* (like turbulent fluids or soft-body plasticity) is premature and even distracting, for a clear indication of their capabilities; certainly so when focusing on Robot Learning and Newtonian Mechanics, like we do. To make sure we are in line with expectations, we change our title from *Benchmarking Physical Reasoning of Video Generative Models with Real Physical experiments* to *Benchmarking Physical Reasoning of Video Generative Models with Real Physical experiments on Newtonian Mechanics.*
>
> Morpheus was carefully designed to be modular, and can thus generalize easily. The co-operation between the trajectory extraction and the scoring components of our framework enable the extension to more complex mechanics with simple additions. This is due to the fact that with this work we create a benchmark, and we can control the experiment settings, allowing for describing analytically the underlying equations of motions.
>
> At the same time we describe how to create more diverse visual augmentations (Section 3 in the main paper), which extend the dataset to visually diverse and physically realistic augmentations, while not disturbing the tracking or the scoring components. This proves the transferability of our approach. In Appendix (Figure 22) we further assess robustness and conclude that the pipeline is robust also with physical reasoning on the style-transfer split.
>
> Considering even more complex physical settings where analytical solutions are intractable (e.g., fluid dynamics), while not the focus of the work, can also be accommodated by Morpheus. We note that Morpheus is perfectly compatible with state-of-the-art physics simulators. As such, in this case we can:
> * Replace analytical (exact) equations by numerical, yet still exact methods, that relevant simulators can provide.
> * Replace real-world video initializations with simulated data, adapted for realism in a Sim2Real fashion.
>
> We strongly believe that the current setup serves as the necessary **first line of work** for quantifying physical reasoning. Attempting to account for complex, stochastic phenomena before models have mastered *de-facto* physics abilities - which we prove they have not - would be premature. Morpheus goes beyond previous approaches by providing an objective, physics-interpretable, and reference-free metric, establishing the blueprint for how physical reasoning should be approached.

---

> ### Author Response · Authors · 2025-11-28
>
> **W2: Metric Robustness to Tracking and Occlusions**
>
> **Reviewer's Comment:** *Noise, occlusions, or tracking drift could propagate into PINN fitting... there is no sensitivity analysis...*
>
> **Our Response:** We thank the reviewer for highlighting this critical aspect of our pipeline. Our pipeline is carefully designed for the robustness of our metrics with the following considerations:
> 1. **Empirical Validation:** If tracking noise or drift were a major source of error, the PINNs would fail to fit the real-world trajectories, leading to low Invariance Scores. Instead, real-world videos achieve near-perfect scores, confirming the robustness of our tracking pipeline: as shown in Figure 8, their scores average 0.93.
> 2. **Handling Occlusions:** We use image conditioning from real-world videos as initialization for VGMs. There the tracked object is never occluded, and the camera is fixed, we can safely assume that the target object is always visible, as confirmed by close to zero discard rates on real-world videos. Therefore, if the object disappears in a generated video (as detected by the tracker), we treat this as non-physical behavior and discard that sample with corresponding minimal scores.
>
> **W3: Manual Design vs. Data-Driven Discovery**
>
> **Reviewer's Comment:** *The design of invariants and ODE priors is manually specified... Integrating data-driven symbolic discovery... could make the benchmark more generalizable.*
>
> **Our Response:** Data-driven discovery of the underlying physics is indeed another way to design and validate a metric based on adherence to the equations of motion. While we agree that data-driven symbolic discovery is a powerful research direction, we believe it introduces significant risks when used as part of Morpheus *benchmark*. Symbolic discovery algorithms are often sensitive to noise, need large amounts of data, and make it hard to ensure a good fit across all governing equations. Thus, a low score might reflect a failure of the discovery algorithm rather than of the video model. For benchmarking, we therefore argue that manually specifying the physics laws is a safer way to obtain a reliable evaluation of physical plausibility.
>
> To address scalability, we suggest a middle ground for future work: a *"library of laws"* approach. Instead of deriving equations manually for every new setup, a small classifier (or VLM) would choose the appropriate governing equations from a predefined library of physics residuals, each tailored to a class of experiments. This preserves the rigor of known physics while scaling to new scenarios without the cost and uncertainty of full symbolic discovery.
>
> **W4: Prompt Engineering and Fairness**
>
> **Reviewer's Comment:** *It is unclear whether performance differences arise from physical reasoning capability or prompt quality. The fairness of cross-model comparisons... remains questionable.*
>
> **Our Response:**
>
> We address the role of the prompt enhancing in the analysis in Section 5 of the revised manuscript (lines 420-424) and further support it with ablation studies in Figure 23 (Appendix, p. 33, highlighted), which quantify the gains in physical reasoning achieved by using prompt upsamplers versus plain prompts.
>
> We agree that direct cross-model comparisons between keyframe interpolation and open-ended generation can be misleading. Our goal, however, is to study different usage regimes in which VGMs can serve as world models, including open-ended generation (image and video) and interpolation (e.g., when a target final frame is given). To this end, we evaluate models in all regimes officially supported by their authors, not to claim strict comparability, but to showcase what each model can achieve under its intended usage. In the final version of the paper, we will explicitly separate keyframe-interpolation and video-conditioned models in the analysis and presentation to make clear that these regimes are not fully comparable, but rather complementary views of model capabilities.

---

> ### Author Response · Authors · 2025-11-28
>
> **Specific Questions**
>
> **Q1: Sensitivity to tracking/depth errors?**
> **Response:** As detailed in W2, the high scores obtained on real-world videos provide empirical upper-bound proof of the pipeline's robustness.
>
> **Q2: PINN hyperparameters ($\lambda$) and comparability?**
> **Response:** For all PINNs, we use fixed, phenomenon-specific $\lambda$ values that are never tuned per generative model, ensuring fair comparability. For simpler systems (freefall, pendulum, spring, projectile, bouncing ball, sliding object), we set $\lambda = 1$, so the total loss is the sum of data and physics terms. For the more chaotic double pendulum, we use $\lambda = 5$ to stabilize training and reduce overfitting to noise. Because all evaluation metrics are computed with the same $\lambda$ for a given phenomenon, performance differences reflect properties of the generated trajectories.
>
> **Q3: 3D Consistency (parallax, occlusion)?**
> **Response:** Morpheus is used to evaluate physical reasoning and operates in a controlled regime where 3D depth ambiguities are minimized. All real-world experiments were recorded using a static, orthogonal camera, and before computing the metrics. We have conducted a depth consistency using both Depth Anything V2 [1] and also Depth Anything V3 [2]. Both DA2 [3] and DA3 [4] indicate that the depth remains essentially constant over the course of the whole sequence. This confirms there is no significant motion along the optical axis, which means that the objects do not move toward or away from the camera. Under this setup, the 2D trajectory in the image plane is a projection of the underlying physical motion, making parallax and occlusion effects negligible. Although Morpheus does not explicitly score 3D consistency (e.g., parallax correctness or dynamics of occlusion), it enforces 3D coherence implicitly for reliable physical evaluation in this more simplified regime. We have also added a dedicated subsection in Appendix D.3 describing this depth consistency check in detail.
>
> **Q4: Extension to non-Newtonian systems?**
> **Response:** Morpheus was carefully designed to be modular, and can thus generalize easily. The co-operation between the trajectory extraction and the scoring components of our framework enables the extension to more complex mechanics with simple additions. This is because with this work, we create a benchmark, and we can control the experiment settings, allowing us to analytically describe the underlying equations of motion.
>
> At the same time, we describe how to create more diverse visual augmentations (Section 3 in the main paper), which extend the dataset to visually diverse and physically realistic augmentations, while not disturbing the tracking or the scoring components. This proves the transferability of our approach. In Appendix (Figure 22), we further assess robustness and conclude that the pipeline is robust, also with physical reasoning on the style-transfer split.
>
> Considering even more complex physical settings where analytical solutions are intractable (e.g., fluid dynamics), while not the focus of the work, can also be accommodated by Morpheus. We note that Morpheus is perfectly compatible with state-of-the-art physics simulators. As such, in this case, we can:
> * Replace analytical (exact) equations by numerical, yet still exact methods, that relevant simulators can provide.
> * Replace real-world video initializations with simulated data, adapted for realism in a Sim2Real fashion.
>
> That said, we focus on Newtonian mechanics, which is the primary concern for using VGMs as world models for robot learning.
>
> **References**
>
> [1] Yang, Lihe, et al. "Depth anything v2." Advances in Neural Information Processing Systems 37 (2024): 21875-21911.
>
> [2] Lin, Haotong, et al. "Depth Anything 3: Recovering the Visual Space from Any Views." arXiv preprint arXiv:2511.10647 (2025).
>
> [3] Li, Haodong, et al. "DA $^{2} $: Depth Anything in Any Direction." arXiv preprint arXiv:2509.26618 (2025).
>
> [4] Lin, Haotong, et al. "Depth Anything 3: Recovering the Visual Space from Any Views." arXiv preprint arXiv:2511.10647 (2025).

---

### Official Review · Reviewer_MqUN · 2025-10-31

**Soundness:** 3
**Presentation:** 3
**Contribution:** 3
**Rating:** 6
**Confidence:** 4

**Summary:**

This paper introduces Morpheus, a new benchmark designed to evaluate the physical reasoning capabilities of video generative models (VGMs). The authors argue that current evaluation methods, which rely on subjective human judgment or simple trajectory matching, are insufficient for rigorously assessing physical plausibility. Morpheus consists of a dataset of real-world videos capturing nine core physical phenomena governed by Newtonian mechanics (e.g., falling objects, projectile motion, pendulums). The proposed methodology extracts object trajectories from both real and generated videos and assesses their physical plausibility using two novel, physics-informed metrics: a "Dynamical Score," which measures adherence to the governing equations of motion via Physics-Informed Neural Networks (PINNs), and a "Physical Invariance Score," which quantifies the conservation of physical quantities like energy and momentum. The authors evaluate several state-of-the-art VGMs and find that, despite generating aesthetically pleasing videos, they struggle to adhere to these fundamental physical principles, highlighting a significant gap in their world modeling capabilities.

**Strengths:**

The main strength of this work is its shift from qualitative, subjective assessments to a quantitative, physics-informed evaluation framework. By grounding the metrics in fundamental conservation laws and equations of motion, the benchmark provides a more objective and rigorous way to measure the physical reasoning of VGMs. This is a significant step forward for the field. The creation of a new dataset based on controlled laboratory experiments is a valuable contribution. This controlled setting allows for the targeted evaluation of specific physical principles and ensures that the comparisons between models are fair and reproducible, which is often a challenge with in-the-wild video data.

**Weaknesses:**

1.Limited Scope of Physical Phenomena: The benchmark's reliance on trajectory extraction from rigid bodies inherently limits its scope. Many important real-world physical phenomena do not involve a clearly trackable rigid object, such as fluid dynamics (e.g., water flowing, smoke rising), thermodynamics (e.g., ice melting), non-rigid body dynamics, or events like explosions. While the focus on Newtonian mechanics is a reasonable starting point, the claim of "benchmarking physical reasoning" is very broad, whereas the method is confined to a relatively narrow fundamental subset of physics.
2.Simplicity of a Majority of the Physical Scenarios: Most of the experiments focus on the motion of a single simple object (e.g., a falling ball, a sliding book). While these are excellent for isolating variables, they may be too "trivial" for the rapid pace of VGM development. State-of-the-art models may quickly learn to master these simple scenarios, potentially limiting the long-term utility of the benchmark. The benchmark could be strengthened by including more complex multi-object interactions or scenarios where secondary physical effects (e.g., air resistance, complex friction) are more prominent.
3.Lack of Correlation with Human Judgment: The paper rightly critiques the subjectivity of human evaluation. However, it does not provide any analysis of how the Morpheus scores correlate with human perception of physical plausibility. A strong benchmark should ideally produce rankings that are consistent with human intuition. Without a correlation study, it is difficult to know if the proposed metrics might sometimes penalize generations that are perceptually plausible (e.g., due to unmodeled physics like friction) or reward generations that are mathematically sound but visually awkward. This comparison is crucial for validating that the metrics are capturing meaningful aspects of physical realism.
Constraints on Evaluation Modality: The proposed method is best suited for image-to-video or video-to-video generation, where the initial state (position, and sometimes velocity) of the object is clearly defined by the conditioning frame(s). This makes it more challenging to fairly evaluate pure text-to-video models, where the model generates the initial state from scratch, introducing ambiguity that the evaluation framework is not designed to handle. This limits the benchmark's applicability to a subset of VGMs.

**Questions:**

1.Could the authors elaborate on the decision to use Depth Anything V2 for checking depth consistency in videos? Were video-specific depth estimation models considered, and would they potentially offer more temporally consistent results that could benefit the analysis?
2.The discard rates for some models are notably high (e.g., 47% for PyramidalFlow in single-frame mode). This suggests a high rate of catastrophic failures before any fine-grained physics analysis can even be performed. Should this high rate of "unevaluable" generations itself be considered a primary metric for physical reasoning, perhaps weighted more heavily in the final assessment of a model's capabilities?
3.How robust is the evaluation pipeline to potential errors from the upstream object segmentation and tracking modules? Since the entire calculation of velocity and acceleration depends on the accuracy of the extracted centroids from SAM-2, even small tracking errors could be amplified into large errors in the final physics scores. Was any sensitivity analysis performed on this?
4.The paper states that the benchmark is restricted to Newtonian physics and assumes negligible friction and air resistance. However, these forces are always present in real-world videos. How does the framework handle a generated video that plausibly models friction (e.g., an object slowing down correctly), which would violate the ideal conservation of energy assumption? Could this lead to physically realistic videos being unfairly penalized?

**Details Of Ethics Concerns:**

The authors have included an ethics statement. The work is focused on benchmarking publicly available models using a newly collected dataset of inanimate objects in controlled experiments. The research does not involve sensitive data, human subjects, or any apparent negative societal impacts. I have no ethical concerns with this paper.

---

> ### Author Response · Authors · 2025-11-28
>
> We thank Reviewer MqUN for their positive assessment and insightful comments. We are grateful that the reviewer recognized the main strength of our work as *a significant step forward for the field* in shifting from subjective assessments to a quantitative, physics-informed framework for quantifying the reasoning capabilities of video models.
>
> **W1: Limited Scope of Physical Phenomena**
>
> **Reviewer's Comment:** *The benchmark's reliance on trajectory extraction from rigid bodies inherently limits its scope... the method is confined to a relatively narrow fundamental subset of physics.*
>
> **Our Response:** Other reviewers also mentioned the scope of physical phenomena, and we clarify here that our focus on Newtonian dynamics is a deliberate scientific choice. To maintain consistency in responses across reviewers, we summarize our rationale and explain how the framework generalizes beyond its current score.
>
> We acknowledge that we focused on Newtonian mechanics. As detailed in our general comment, even proprietary SOTA models (Kling-2.5-Turbo, Veo3 and Veo3-fast) **fail to consistently solve such *simple* scenarios**, struggling with basic conservation laws like energy and momentum. Until models can master these ABCs of physics, evaluating them on *complicated physics* (like turbulent fluids or soft-body plasticity) is premature and even distracting, for a clear indication of their capabilities; certainly so when focusing on Robot Learning and Newtonian Mechanics, like we do. To make sure we are in line with expectations, we change our title from *Benchmarking Physical Reasoning of Video Generative Models with Real Physical experiments* to *Benchmarking Physical Reasoning of Video Generative Models with Real Physical experiments on Newtonian Mechanics.*
>
> Morpheus was carefully designed to be modular, and can thus generalize easily. The co-operation between the trajectory extraction and the scoring components of our framework enables the extension to more complex mechanics with simple additions. This is because with this work, we create a benchmark, and we can control the experiment settings, allowing us to describe analytically the underlying equations of motion.
>
> At the same time, we describe how to create more diverse visual augmentations (Section 3 in the main paper), which extend the dataset to visually diverse and physically realistic augmentations, while not disturbing the tracking or the scoring components. This proves the transferability of our approach. In Appendix (Figure 22), we further assess robustness and conclude that the pipeline is robust, also with physical reasoning on the style-transfer split.
>
> Considering even more complex physical settings where analytical solutions are intractable (e.g., fluid dynamics), while not the focus of the work, can also be accommodated by Morpheus. We note that Morpheus is perfectly compatible with state-of-the-art physics simulators. As such, in this case, we can:
> * Replace analytical (exact) equations by numerical, yet still exact methods, that relevant simulators can provide.
> * Replace real-world video initializations with simulated data, adapted for realism in a Sim2Real fashion.
>
> We firmly believe that Morpheus serves as the necessary **first line of work** for assessing physical reasoning using physics-informed interpretable metrics. Attempting to account for complex, stochastic phenomena before models have mastered *de facto* physics abilities - which we prove they have not - would be premature. Morpheus goes beyond previous approaches by providing an objective, physics-interpretable, and reference-free metric, establishing the blueprint for how physical reasoning should be approached.

---

> ### Author Response · Authors · 2025-11-28
>
> **W2: Simplicity of a Majority of the Physical Scenarios**
>
> **Reviewer's Comment:** *Most of the experiments focus on the motion of a single simple object... they may be too "trivial" for the rapid pace of VGM development.*
>
> **Our Response:** We thank the reviewer for raising this important point about the benchmark's granularity in supporting a variety of interactions. While several scenarios involve single objects to isolate variables, we would like to highlight experiments like *collisions* and the *double pendulum* that test multi-object interaction and chaotic dynamics, respectively. Either way, even the latest closed-source VGMs are still far from true physical performance, even in single-object scenarios (Figure 7 of the updated paper).
>
> **W3: Lack of Correlation with Human Judgment**
>
> **Reviewer's Comment:** *...it does not provide any analysis of how the Morpheus scores correlate with human perception of physical plausibility.*
>
> **Our Response:**
> Thanks for raising this point. We would like to clarify that Morpheus intentionally prioritizes objective physical quantification in VGMs over the human-preference paradigm extensively explored by prior subjective benchmarks (e.g., VideoPhy [1], PhyGenBench [3]). Our rationale is twofold:
>
> * **Physical Reasoning without Discrete Scores Vision Question Answering Scores:** Our proposed metrics capture adherence to equation of motions and conservation of physical properties rather than bounded discrete scores at some scale (0: being the lowest, 5: being the highest) limited to human judgment of "*qualitatively correct*" physics. Our metrics are not only more objective but rather interpretable to well-known physics equations and conservation laws. The latter alleviates the need for annotating hours of videos one by one with a discrete score subjective to human subjective judgment (Semantic Adherence, Physical Commonsense VQA scores respectively, as defined by VideoPhy [1]).
> * **Alleviate the Human-in-Loop necessity:** Previous work like PhyWorldBench [2], VideoPhy [1], and PhyGenBench [3] suggest an approach that is simply not scalable. Approaches like VideoPhy and PhyWorldBench require expensive finetuning of their VLM-expert models on new data points for every new type of physics they wish to support. These new data points need expensive annotation with Multimodal-LLMs which are then need to filtered by human annotators. By the end of this process to benchmark a new video generation model, a VLM will provide scores, which again have to be tested and correlated to the human judgment. By contrast, Morpheus is designed to require as little human annotation as possible.
>
> Regarding "perceptually plausible" generations: **Figure 8** demonstrates that real-world videos achieve near-perfect scores, confirming that unmodeled factors like friction do not negatively impact our evaluation. Conversely, regarding mathematically sound but "visually awkward" generations: visual artifacts (e.g., flickering, severe deformation) disrupt the tracking pipeline, triggering our discard mechanism and correctly resulting in low scores.
>
> Nevertheless, we agree that it is useful to compare whether human perception of physical plausibility has high correlation with Morpheus score grounded in analysis for the object trajectories. Thus we plan to extend the analysis of Morpheus results with human perception evaluation both for discard rate and overall Morpheus score for camera ready version of the paper, upon acceptance.

---

> ### Author Response · Authors · 2025-11-28
>
> **W4: Constraints on Evaluation Modality**
>
> **Reviewer's Comment:** *The proposed method is best suited for image-to-video or video-to-video generation... This limits the benchmark's applicability to a subset of VGMs.*
>
> **Our Response:** We agree that our current experiments focus on conditioned generation (I2V/V2V). This focus aligns with our goal of evaluating VGMs as potential "world models," where predicting the future from a given current visual observation or observation history is a key task needed for planning or learning in imagination. However, the Morpheus evaluation pipeline itself is not fundamentally limited to those settings. It can be easily adapted to support T2V models (text modality only) by leveraging recent advances in zero-shot open-vocabulary trackers (e.g., Grounded-SAM 2 [6], SAM-3 [7]) to automate the initial object annotation. This is out of the scope of the current work though.
>
> **Q1: Use of Depth Anything V2**
>
> **Reviewer's Question:** *Could the authors elaborate on the decision to use Depth Anything V2... Were video-specific depth estimation models considered?*
>
> **Our Response:**
> We decided to use Depth Anything because depth estimation in our pipeline is only for the purpose of a sanity check, to ensure that the camera assumptions (orthogonal and static) are indeed true. Depth is not used in the Dynamical Score or the Physical Invariance Score. We only use it to confirm that the tracked objects does not move towards or away from the camera.
> In the first submission, we used Depth Anything V2 [4], which gave us stable frame-by-frame depth estimates for our setup. In this revised version, we also run the same check with the recently released video-based Depth Anything V3 [5]. For each video, we calculated the average depth, and we found out that depth remains consistent. Both DA2 [8] and DA3 [9] yield constant depth values for the real-world videos (and also the generated ones). We have added a subsection in the appendix showcasing the results for real-world videos (App. D.3).
>
> **Q2: High Discard Rates as a Primary Metric**
>
> **Reviewer's Question:** *Should this high rate of "unevaluable" generations itself be considered a primary metric... perhaps weighted more heavily?*
>
> **Our Response:** We thank the reviewer for this suggestion, and we wholeheartedly agree. The **discard rate is indeed a crucial, primary metric** of our benchmark, as detailed in Section 4. It quantifies a model's basic reliability and coherence. A model that frequently fails to generate plausible futures (e.g., by having objects disappear or failing to move) is fundamentally flawed, regardless of how well its few successful generations score. We account for discarded cases in the final scores by assigning physical and dynamical scores of 0 to any discarded outputs, which is the minimal possible score. This is consistently reflected in all our evaluations and model rankings. We believe this adequately penalizes discarded outputs, but we are open to discuss concrete alternative suggestions.
>
> **Q3 & Q4: Pipeline Robustness and Handling of Friction**
>
> **Reviewer's Question:** *How robust is the evaluation pipeline to potential errors from... tracking modules? ...How does the framework handle a generated video that plausibly models friction...?*
>
> **Our Response:**
>
> **Robustness to Tracking Errors.** The strongest evidence for our pipeline's robustness is the fact that \textbf{real-world videos achieve near-perfect scores}. This outcome would be impossible if the underlying SOTA tracking from SAM-2 were introducing significant errors. We find that tracking failures are almost always caused by implausible generations from the VGM itself (e.g., object disappearance), which our discard mechanism correctly penalizes.
>
> **Handling of Plausible Friction**
> Similarly, the near-perfect scores on real-world videos empirically demonstrate that effects like friction and air resistance are negligible in our controlled settings. This validates that our ideal-physics-based framework does not unfairly penalize plausible, real-world motion and serves as a robust baseline for evaluation.

---

> > ### Author Response · Authors · 2025-12-03
> >
> > **References**
> >
> > [1] Bansal, Hritik, et al. "Videophy: Evaluating physical commonsense for video generation." arXiv preprint arXiv:2406.03520 (2024).
> >
> > [2] Gu, Jing, et al. "" PhyWorldBench": A Comprehensive Evaluation of Physical Realism in Text-to-Video Models." arXiv preprint arXiv:2507.13428 (2025).
> >
> > [3] Meng, Fanqing, et al. "Towards world simulator: Crafting physical commonsense-based benchmark for video generation." arXiv preprint arXiv:2410.05363 (2024).
> >
> > [4] Yang, Lihe, et al. "Depth anything v2." Advances in Neural Information Processing Systems 37 (2024): 21875-21911.
> >
> > [5] Lin, Haotong, et al. "Depth Anything 3: Recovering the Visual Space from Any Views." arXiv preprint arXiv:2511.10647 (2025).
> >
> > [6] Ren, Tianhe, et al. "Grounded sam: Assembling open-world models for diverse visual tasks." arXiv preprint arXiv:2401.14159 (2024).
> >
> > [7] Carion, Nicolas, et al. "SAM 3: Segment Anything with Concepts." arXiv preprint arXiv:2511.16719 (2025).
> >
> > [8] Li, Haodong, et al. "DA $^{2} $: Depth Anything in Any Direction." arXiv preprint arXiv:2509.26618 (2025).
> >
> > [9] Lin, Haotong, et al. "Depth Anything 3: Recovering the Visual Space from Any Views." arXiv preprint arXiv:2511.10647 (2025).

---

### Official Review · Reviewer_Z6tA · 2025-10-31

**Soundness:** 3
**Presentation:** 3
**Contribution:** 2
**Rating:** 4
**Confidence:** 4

**Summary:**

Summary:
This paper addresses the critical limitation of current video generative models (VGMs): their inability to adhere to fundamental physical laws despite producing visually realistic content. To solve this, the authors introduceMorpheus, a novel benchmark for evaluating VGMs’ physical reasoning capabilities using real-world physical experiments.

Contributions:
（1）Innovative quantitative evaluation framework: The combination of Dynamical Score (PINN-based equation fitting) and Physical Invariance Score (invariant conservation) provides objective, fine-grained metrics for physical plausibility, resolving the limitations of prior methods that fail to quantify subtle physical violations.
（2）Extensive VGM evaluation and actionable insights: The large-scale evaluation (6 models, 9000 videos) identifies critical gaps and reveals that multi-frame conditioning and prompt enhancement modestly improve physical realism—guiding future VGM optimization.
（3）Visually diverse conditioning for robust evaluation: Morpheus augments real-world videos with style transfers, ensuring evaluations are robust to visual variations in VGM training data.

**Strengths:**

Strengths:
（1）Originality
Morpheus, its proposed benchmark, is the first to use controlled real-world physics experiments for VGM physical reasoning evaluation. It designs a dual-metric (Dynamical + Physical Invariance Score) framework with PINNs for objective physical plausibility measurement. It also uses style transfers for diverse conditioning, ensuring robust evaluations.
（2）Quality
Its dataset comes from 9 controlled experiments (varied initial conditions) and metrics are validated with real videos. It evaluates 6 VGMs across 9000 videos, with systematic analyses of prompts and conditioning modes. Ablations isolate variable impacts to avoid cherry-picking.
（3）Clarity
It follows a clear “gap→solution→validation” flow, explaining complex concepts simply. Consistent terms and visual aids help readers follow design and results easily.
（4）Significance
As an open tool, it provides a shared platform for tracking VGM progress, guiding real-world uses. It shifts VGM evaluation to prioritize physical correctness and aligns with scientific integrity.

**Weaknesses:**

Weaknesses:
（1）Narrow Physics Scope
Morpheus only covers Newtonian physics in controlled lab settings, excluding domains like fluid dynamics or electromagnetism. It assumes negligible air resistance/friction, which may unfairly penalize plausible generations accounting for these real-world factors.
（2）Uncalibrated 2D Trajectory Extraction
It uses 2D pixel trajectories without camera calibration to real-world units, leading to ambiguous measurements. This risks inaccurate Dynamical/Physical Invariance Scores, especially for experiments needing precise distance/velocity data.
（3）The comparison with the methods of predecessors is not sufficient.
A comparison between the dataset and some previous related datasets, such as VideoREPA, WISA, NewtonGen, T2vphysbench, etc.?
（4）Missing Closed-Source VGM Evaluation
It excludes closed-source models (e.g., SORA) due to budget limits, omitting state-of-the-art performance. This weakens benchmark comprehensiveness, as readers can’t assess the full VGM spectrum.
（5）Biased Time Window for Scores
Physical Invariance Scores use different time window sizes for real (10% duration) and generated (25% duration) videos. No justification is given, and this inconsistency may artificially skew scores.

**Questions:**

（1）Physics Scope Limitation
Your benchmark Morpheus only covers Newtonian physics in controlled laboratory settings and excludes domains like fluid dynamics or electromagnetism. Do you have plans to expand Morpheus to include these broader physical domains in future updates? Additionally, since you assume negligible air resistance and friction, how would you adjust the evaluation metrics to avoid penalizing physically plausible generations that account for these real-world factors?
（2）2D Trajectory Extraction
Morpheus uses 2D pixel trajectories without camera calibration to convert pixels into real-world physical units (e.g., meters, seconds). Have you tested whether adding camera calibration would reduce ambiguity in Dynamical and Physical Invariance Scores, especially for experiments requiring precise distance or velocity measurements? If not, what challenges prevent implementing such calibration?
（3）Closed-Source VGM Inclusion
You exclude closed-source VGMs (e.g., SORA) due to budget limits, which weakens the benchmark’s comprehensiveness. We suggest partnering with providers of closed-source models for limited access, or using publicly available generated videos from these models (if accessible) to include them in future evaluations. This would allow readers to assess how state-of-the-art closed-source models perform against Morpheus’ metrics.
If the author can effectively solve my doubts, I will consider improving my score.

---

> ### Author Response · Authors · 2025-11-28
>
> We sincerely thank Reviewer Z6tA for their thorough and constructive feedback. We are pleased that the reviewer appreciated the originality and significance of our work as a step toward an objective, physics-informed evaluation of VGMs. We take the comments as an opportunity to strengthen the relevance of our work. Below, we address each weakness and answer each question.
>
> **W1 & Q1: Narrow Physics Scope and Handling of Real-World Factors**
>
> **Reviewer's Comment:** *Morpheus only covers Newtonian physics... excluding domains like fluid dynamics... how would you adjust the evaluation metrics to avoid penalizing physically plausible generations that account for these real-world factors (like friction)?*
>
> **Our Response:** The scope of physical phenomena was also raised by other reviewers and we clarify here that our focus on Newtonian dynamics is a deliberate scientific choice. To keep the response consistent across reviewers, we summarize our rationale and how the framework generalizes beyond its current score.
>
> We acknowledge that we focused on Newtonian mechanics. As detailed in our general comment, even SOTA models (Google's Veo family models) **fail to consistently solve such *simple* scenarios**, struggling with basic conservation laws like energy and momentum. Until models can master these ABCs of physics, evaluating them on *complicated physics* (like turbulent fluids or soft-body plasticity) is premature and even distracting, for a clear indication of their capabilities; certainly so when focusing on Robot Learning and Newtonian Mechanics, like we do. To make sure we are in line with expectations, we change our title from *Benchmarking Physical Reasoning of Video Generative Models with Real Physical experiments* to *Benchmarking Physical Reasoning of Video Generative Models with Real Physical experiments on Newtonian Mechanics.*
>
> Morpheus was carefully designed to be modular, and can thus generalize easily. The co-operation between the trajectory extraction and the scoring components of our framework enables the extension to more complex mechanics with simple additions. This is due to the fact that with this work, we create a benchmark, and we can control the experiment settings, allowing us to describe analytically the underlying equations of motion.
>
> At the same time, we describe how to create more diverse visual augmentations (Section 3 in the main paper), which extend the dataset to visually diverse and physically realistic augmentations, while not disturbing the tracking or the scoring components. This proves the transferability of our approach. In Appendix (Figure 22), we further assess robustness and conclude that the pipeline is robust, also with physical reasoning on the style-transfer split.
>
> Considering even more complex physical settings where analytical solutions are intractable (e.g., fluid dynamics), while not the focus of the work, can also be accommodated by Morpheus. We note that Morpheus is perfectly compatible with state-of-the-art physics simulators. As such, in this case, we can:
> * Replace analytical (exact) equations by numerical, yet still exact methods, that relevant simulators can provide.
> * Replace real-world video initializations with simulated data, adapted for realism in a Sim2Real fashion.
>
> That said, we focus on Newtonian mechanics, which is the primary concern for using VGMs as world models for robot learning.
>
> Regarding factors for which our pipeline does not model (like friction), our comprehensive experiments with real-world videos provide a good indication of whether it is fair to consider such quantities negligible. We point the reviewer to Figure 8 of our manuscript. **The real-world videos in our benchmark achieve near-perfect scores** on both the Dynamical and Physical Invariance metrics. This empirically showcases that for our real-world experiments, ideal Newtonian laws are a strong and good approximation, and the effects of factors like air resistance and friction are indeed negligible.

---

> ### Author Response · Authors · 2025-11-28
>
> **W2 & Q2: 2D Trajectory Extraction: Calibration and Robustness**
>
> **Reviewer's Comment:** *Morpheus uses 2D pixel trajectories without camera calibration... Have you tested whether adding camera calibration would reduce ambiguity?*
>
> **Our Response:** Thanks for posing such an important technical question. We address it in two parts: camera calibration and the general robustness of our tracking pipeline.
>
> **1. Invariance to Camera Calibration:**
> To address the use of 2D pixel trajectories without explicit camera calibration, we empirically tested the scale robustness of both the tracking pipeline and the PINN-based Dynamical Score. We rescaled each trajectory by factors from 0.5× to 2×, and the Dynamical Score remained very close to 1 throughout. In the freefall experiment, scores stayed between 0.96 and 1.00, and for the pendulum, between 0.97 and 1.00, indicating that the PINN fit is effectively preserved under rescaling. This shows that the PINN reliably recovers the underlying dynamics over a broad range of pixel scales, so moderate changes in camera distance or object size do not affect the Dynamical Score. For the physical invariance score, we use additional scaling to transform pixel coordinates to world coordinates. We have included an additional subsection in the appendix discussing more about this and showcasing the aforementioned results (Appendix D.4).
>
> **2. Robustness of Trajectory Extraction:** The reliability of our entire evaluation framework rests on accurate trajectory extraction. We validate this directly: the near-perfect scores achieved on real-world videos are only possible if the underlying tracking from SOTA models like SAM-2 is highly accurate for our dataset. Unlike opaque VLM-based scores, the extracted trajectories from Morpheus can be visually inspected, providing a transparent and verifiable foundation for our metrics.
>
> **W3: Comparison with Predecessors**
>
> **Reviewer's Comment:** *The comparison with the methods of predecessors is not sufficient...*
>
> **Our Response:**
> We sincerely thank the reviewer for making us think about the positioning of our paper. Consequently, we have updated the **Physical Reasoning and plausibility in VGMs** (Related Work section - lines 150-156), where we cover the limitations of existing works, in terms of designing benchmarks useful for the evaluation of VGMs as a foundation for physically plausible world models.
>
> WISA[1] introduces a dataset (WISA-32k) which includes physically-rich annotations (that capture qualitative categories and quantitative physical properties) employing multi-modal VLMs by decomposing physical information from 32,000 videos scraped from the internet. Yet the evaluation of their work is carried out by employing VLM-based proxy scores along with another human correlation study. Closer to our work, T2VPhysBench[2] is a human evaluation protocol that covers the adherence of video models to capturing Newtonian mechanics, conservation principles, and phenomenological effects. VideoREPA[3] does not introduce any new dataset, and its evaluation likewise employs VLM-based metrics and human-correlation studies. NewtonGen[4] is a successor work that directly draws inspiration from Morpheus' physical invariance scores, used in their evaluation. The key difference between Morpheus and the afore-mentioned line of works is that we prioritize physical reasoning required for manipulation tasks, particularly relevant in **Robot Learning**.

---

> ### Author Response · Authors · 2025-11-28
>
> **W4 & Q3: Missing Closed-Source VGM Evaluation**
>
> **Reviewer's Comment:** *It excludes closed-source models (e.g., SORA)... This weakens benchmark comprehensiveness.*
>
> **Our Response:** We thank the reviewer for this suggestion. We have expanded our benchmark to include two dedicated closed-source models SOTA models (Kling-2.5-Turbo, Veo-3), with two different variants of Veo-3 belonging to the same Google's family model. We have included them in Figure 7 and we discuss their results in the paper (lines 431-450). As expected, they outperform other open-source models; however there is still a significant gap to the real-world videos' performance (0.60 vs 0.93).
>
> As to make sure for **MORPHEUS** to stay relevant, we will do our best to keep expanding our leaderboard to more closed-source SOTA models. We have contacted the teams behind Ray3 (Luma Labs), Gen-4 / Gen-4 Turbo (Runway), Pika 2.2 (Pika Labs), PixVerse V5, and Veo 3.1 (Google) to request research access and API credits for evaluation.
>
> As promised we have delivered Kling's family video models (Kling-2.5-Turbo) and hosted them into our leaderboard, since the last iteration.
>
> **W5: Biased Time Window for Scores**
>
> **Reviewer's Comment:** *Physical Invariance Scores use different time window sizes for real (10% duration) and generated (25% duration) videos...*
>
> **Our Response:** We thank the reviewer for highlighting this design detail. The original choice was made to keep the absolute window lengths similar, since real-world videos were often longer. To avoid confusion, we have now standardized video length and all window sizes. This change has only a minor impact on Morpheus scores for real videos (1–3% difference as can be seen in updated Figure 8) and does not affect our main conclusion: there remains a large gap between the best VGMs and real-world videos (0.45 vs. 0.93).
>
> **References**
>
> [1] Wang, Jing, et al. "Wisa: World simulator assistant for physics-aware text-to-video generation." arXiv preprint arXiv:2503.08153 (2025).
>
> [2] Guo, Xuyang, et al. "T2vphysbench: A first-principles benchmark for physical consistency in text-to-video generation." arXiv preprint arXiv:2505.00337 (2025).
>
> [3] Zhang, Xiangdong, et al. "VideoREPA: Learning Physics for Video Generation through Relational Alignment with Foundation Models." arXiv preprint arXiv:2505.23656 (2025).
>
> [4] Yuan, Yu, et al. "NewtonGen: Physics-Consistent and Controllable Text-to-Video Generation via Neural Newtonian Dynamics." arXiv preprint arXiv:2509.21309 (2025).
>
> **Links to models**
>
> 1. **Ray3 (Luma Labs)**
>    - Website: [https://lumalabs.ai](https://lumalabs.ai)
>
> 2. **Kling 2.5 Turbo (Kuaishou)**
>    - Website: [https://klingai.com](https://klingai.com)
>
> 3. **Gen-4 / Gen-4 Turbo (Runway)**
>    - Website: [https://runwayml.com](https://runwayml.com)
>
> 4. **Pika 2.2 (Pika Labs)**
>    - Website: [https://pika.art](https://pika.art)
>
> 5. **PixVerse V5**
>    - Website: [https://pixverse.ai](https://pixverse.ai)
>
> 6. **Veo 3.1 (Google)**
>    - Website: [https://deepmind.google/technologies/veo/](https://deepmind.google/technologies/veo/)

---

### Official Review · Reviewer_AAtS · 2025-11-03

**Soundness:** 2
**Presentation:** 3
**Contribution:** 2
**Rating:** 2
**Confidence:** 4

**Summary:**

The paper studies the physics evaluation of video generation models. Current evaluation methods rely on subjective judgements or trajectory matching. This paper introduces a new benchmark on physical reasoning, which features 130 real world videos capturing physical phenomena guided by conservation laws. The paper shows that even with advanced prompting and video conditioning, the models still cannot encode physical principles very well

**Strengths:**

1. The paper introduces a benchmark to systemically evaluate physical reasoning of VGMs based on physical invariants using real world physical experiments.
2. The paper is clear and easy to read.

**Weaknesses:**

1. The benchmark only covers limited scenarios and has strong assumption of the environment. However, there are far more videos that might compose more complicated physics.
2. The evaluation framework strongly relies on several other models like sam2, depth anything v2. I do not know in practice, if the framework can be generalized since each part can have some errors, which can be accumulated to influence the reliability of the final prediction of the morpheus score.
3. It does not include the latest SOTA VGMs like veo 3.1, sora 2, etc.

**Questions:**

1. How can this framework be generalized to other scenarios? We actually expect the model to generate very complicated physics dynamics now.
2. What is the performance on the latest close source VGMs?

---

> ### Author Response · Authors · 2025-11-28
>
> We thank Reviewer AAtS for their assessment of our work. We understand that the reviewer's primary hesitation stems from the scope of physics covered and the reliance on open-source models only. We believe that the inclusion of new experimental results (the current version includes two Veo3 variants) and our clarifications on the scope and generalization of Morpheus, address these concerns.
>
> **W1 & Q1: Scope of Scenarios and Generalization**
>
> **Reviewer's Comment:** *The benchmark only covers limited scenarios... We actually expect the model to generate very complicated physics dynamics now.*
>
> **Our Response:** We acknowledge that we focused on Newtonian mechanics. As detailed in our general comment, even SOTA models (Google's Veo family models) **fail to consistently solve such *simple* scenarios**, struggling with basic conservation laws like energy and momentum. Until models can master these ABCs of physics, evaluating them on *complicated physics* (like turbulent fluids or soft-body plasticity) is premature and even distracting, for a clear indication of their capabilities; certainly so when focusing on Robot Learning and Newtonian Mechanics, like we do. To make sure we are in line with expectations, we change our title from *Benchmarking Physical Reasoning of Video Generative Models with Real Physical experiments* to *Benchmarking Physical Reasoning of Video Generative Models with Real Physical experiments on Newtonian Mechanics.*
>
> Regarding the comment on the generalization to new scenarios, Morpheus was carefully designed to be modular, and can thus generalize easily. The co-operation between the trajectory extraction and the scoring components of our framework enables the extension to more complex physics dynamics with simple additions. This is due to the fact that with this work, we create a benchmark, and we can control the experiment settings, allowing for an analytical description of the underlying equations of motion.
>
> At the same time, we describe how to create more diverse visual augmentations (Section 3 in the main paper), which extend the dataset to visually diverse and physically realistic augmentations, while not disturbing the tracking or the scoring components. This proves the transferability of our approach. In Appendix (Figure 22), we further assess robustness and conclude that the pipeline is robust, also with physical reasoning on the style-transfer split.
>
> Considering even more complex physical settings where analytical solutions are intractable (e.g., fluid dynamics), while not the focus of the work, can also be accommodated by Morpheus. We note that Morpheus is perfectly compatible with state-of-the-art physics simulators. As such, in this case we can:
> * Replace analytical (exact) equations by numerical, yet still exact methods, that relevant simulators can provide.
> * Replace real-world video initializations with simulated data, adapted for realism in a Sim2Real fashion.
>
> That said, we focus on Newtonian mechanics, which is the primary concern for using VGMs as world models for robot learning.
>
> **W2: Reliability and Error Accumulation from the tracking pipeline**
>
> **Reviewer's Comment:** *The evaluation framework strongly relies on... sam2, depth anything v2... errors can be accumulated to influence the reliability.*
>
> **Our Response:** We agree it is important to analyze the effects of error accumulation. We evaluate the tracking pipeline on a large number of real-world videos. If the tracking pipeline suffered from substantial error accumulation, it would fail to track objects and fit their dynamics to such data. In contrast, our metrics consistently reach near-perfect scores (i.e., zero discard rates, Dynamical Score ~ 0.96 and Invariance Score ~ 0.90; see Fig. 8). Morpheus' strong performance on real videos therefore shows that the pipeline is both precise and reliable.
>
> **W3 & Q2: Inclusion of Latest Closed-Source VGMs**
>
> **Reviewer's Comment:** *It does not include the latest SOTA VGMs like veo 3.1, sora 2... What is the performance on the latest close source VGMs?*
>
> **Our Response:**
> The updated manuscript now contains a comparison with Kling-2.5-Turbo [2], Veo3 [6] and Veo3-fast [6] models (see Fig. 7 along with highlighted comments lines 431-450).
>
> As to make sure for **MORPHEUS** to stay relevant, we will do our best to keep expanding our leaderboard to more closed-source SOTA models. We have contacted the teams behind Ray3 [1], Gen-4 / Gen-4 Turbo [3], Pika 2.2 [4], PixVerse V5 [5], and Veo 3.1 [6] to request research access and API credits so that their models can be evaluated and included into our benchmark.

---

> ### Author Response · Authors · 2025-11-28
>
> **References**
> 1. **Ray3 (Luma Labs)**
>    - Website: [https://lumalabs.ai](https://lumalabs.ai)
>
> 2. **Kling 2.5 Turbo (Kuaishou)**
>    - Website: [https://klingai.com](https://klingai.com)
>
> 3. **Gen-4 / Gen-4 Turbo (Runway)**
>    - Website: [https://runwayml.com](https://runwayml.com)
>
> 4. **Pika 2.2 (Pika Labs)**
>    - Website: [https://pika.art](https://pika.art)
>
> 5. **PixVerse V5**
>    - Website: [https://pixverse.ai](https://pixverse.ai)
>
> 6. **Veo 3.1 (Google)**
>    - Website: [https://deepmind.google/technologies/veo/](https://deepmind.google/technologies/veo/)

---

### Author Response · Authors · 2025-11-28
**General Response**

**General Response: Depth, Rigor, and the Necessity of our Benchmark**

We thank all reviewers for their insightful feedback. We are encouraged that reviewers recognized the value of grounding VGM evaluation in controlled, real-world physical experiments (Reviewer AAtS, Z6tA, MqUN, LyJG) and highlighted the significance of moving from subjective assessments towards a rigorous, quantitative evaluation framework based on physics-informed metrics (Reviewer Z6tA, MqUN, LyJG).

### Benchmarking with the latest SOTA VGMs

Reviewers AAtS and Z6tA rightly pointed out the importance of evaluating our benchmark on state-of-the-art, closed-source models. We fully agree that including such models is crucial to demonstrate the relevance of our method to the latest advances in the field. In the revised version of the paper, we have included two closed-source state-of-the-art. Kling-2.5-Turbo [5] and Veo3(Veo3 and Veo3-fast variant)[6].

Overall, as can be seen from the newly added scores corresponding to closed-sourced models, while they do perform better than the open-source counterparts. Specifically even for the best performing model (Kling-2.5-Turbo) there remains a significant gap (0.60 vs 0.93) to real-world videos, proving that **Morpheus** is far from saturated. The aggregated results across all physical experiments (Fig. 7 in our manuscript) indicate the relevance of our approach for benchmarking contemporary best-performing video models. Our findings reinforce our argument regarding the novelty of our physical reasoning quantification and the robustness of the design of our physics scenarios that test the limits of video models.

As to make sure that our leadboard stays relevant as up-to-date as possible, we have contacted teams behind the main closed-source VGMs, such as Luma Labs (Ray3), Runway (Gen4 & Gen4 Turbo), Pika Labs (Pika 2.2), PixVerse (V5) and Google (Veo3.1) and to hosting platforms such as Pollo.ai, Vadoo.tv, and rundiffusion.com. Whenever we granted access to these models, we would include them in our leadboard.

### Objective Quantification for Robot Learning

A shared concern across reviewers related to the scope of physical phenomena covered in our work. We would like to clarify the literature gap **Morpheus** tries to cover. Existing benchmarks like VideoPhy [1], VideoPhy2 [2], and PhyGenBench [3] prioritize breadth (e.g. fluids, deformable bodies, optics), relying on subjective VQA scores provided by VLM and then testing the alignment with human preferences. We are happy to see that the reviewers also acknowledge these points. In contrast, **Morpheus** is the first objective work in physical reasoning, prioritizing various Newtonian mechanics (rigid-body dynamics, collisions, chaotic systems, periodic oscillations), as this constitutes the most fundamental layer of reasoning that needs to be mastered for object manipulation and planning, relevant in the direction of using VGMs as world models in the robot learning literature. By utilizing image and video conditioning, one can define the necessary environment initializations, ensuring that our pipeline focuses on accurate transition dynamics rather than the interpretation of potentially ambiguous text prompts.

---

> ### Author Response · Authors · 2025-11-28
> **General Response pt.2**
>
> ### The Ingenuity of Invariance
>
> By contrast to subjective, language-based evaluation, there exist benchmarks that attempt quantitative evaluation via trajectory matching against a single reference video. These lines of works assume there is only one correct future, since all metrics are reported with respect to some ground-truth video. However, in physical generation small changes might generate nearly valid—if not equivalent—futures. A different friction coefficient, or air resistance, can lead to different videos, although the initial frames will invariably look very similar. Even more so, when generating chaotic phenomena such as a double pendulum, no two trajectories are the same even when starting by the same initial frames.
>
> As such, benchmarks like PhysicsIQ [4] unfairly penalize valid variations, as they are agnostic to multiple feasible physics demonstrations.
>
> Our framework introduces the **Dynamical Score** (via PINNs) and the **Physical Invariance Score**. We do not ask *"Does the pixel match the ground truth?"* but rather *"Does the generation follow the equation of motion? Does it conserve Momentum?"* This allows the model to generate *any* plausible future that adheres to the governing equations. This represents a fundamental shift in how physical reasoning is approached by checking adherence to laws, not merely memorization of pixels.
>
> ### A Necessary Stepping Stone
>
> The proposed type of physics in our submission constitute the "ABCs" of Newtonian mechanics. Mastering them is a seminal step before video generative models are deployed, especially when going beyond simple dynamics or stochastic phenomena. Although we start from Newtonian mechanics, our benchmark is inherently modular to handle more complex physics such as fluid dynamics, by replacing the analytical equation (in our dynamical score calculation) with numerical estimations from simulators, and with closing the gap between simulation to real data with Sim2Real transfer adaptation (similarity to our Physically realistic augmentations). **Morpheus** established a first-principle physical quantification method, with metrics connected to physical quantities.
>
> In summary, we explicitly sacrifice the ``illusion of breadth'' to provide an objective, equation-informed, and reference-free metric for physical reasoning in video generation, with a focus on downstream tasks relevant to the usage of VGMs as a foundation for physically plausible world models.
>
> ***
>
> **References:**
>
>
> [1] Bansal, Hritik, et al. "Videophy: Evaluating physical commonsense for video generation." arXiv preprint arXiv:2406.03520 (2024).
>
> [2] Bansal, Hritik, et al. "Videophy-2: A challenging action-centric physical commonsense evaluation in video generation." arXiv preprint arXiv:2503.06800 (2025).
>
> [3] Meng, Fanqing, et al. "Towards world simulator: Crafting physical commonsense-based benchmark for video generation." arXiv preprint arXiv:2410.05363 (2024).
>
> [4] Motamed, Saman, et al. "Do generative video models understand physical principles?." arXiv preprint arXiv:2501.09038 (2025).
>
> [5] Veo 3.1 (Google)
>    - Website: [https://deepmind.google/technologies/veo/](https://deepmind.google/technologies/veo/)
>
> [6] Kling-2.5-Turbo (Luma Labs)
>    - Website: [https://klingai.com](https://klingai.com)

---

### Author Response · Authors · 2025-12-03
**Final comments**

We sincerely thank the reviewers for their insightful and constructive feedback. Your comments helped us clarify the aim of the benchmark, provide more up-to-date empirical results, and highlight the broader applicability of our approach.

In response to the reviewers’ suggestions, we have introduced the following improvements:
 - **Expanded evaluation with state-of-the-art VGMs.** We added results for Veo-3, Veo-3-fast, and Kling-2.5-Turbo, showing that even leading state-of-the-art models perform far below perfect physical accuracy (0.6 vs. 0.93). These new results reinforce that Morpheus remains unsaturated and continues to expose meaningful differences between models.
 - **Robustness checks for calibration and temporal settings.** We confirmed experimentally that the benchmark’s scores are insensitive to camera calibration errors and stable across varying time window lengths, addressing concerns about potential confounding factors. The near-perfect scores of real-world videos further confirm the consistency of the trajectory extraction pipeline.
 - **Discussion of broader physical domains.** In response to the questions about extensibility, we clarified the paper’s scope and its importance for using VGMs as foundations for world models, and we described how this scope could be expanded in future work.
 - **Text revisions.** We substantially extended the related works section, improving clarity and ensuring that the benchmark is accurately positioned relative to prior work.

We hope these revisions fully address the reviewers’ concerns and help clarify the contribution and long-term utility of our benchmark. Thank you again for the thoughtful feedback and for the opportunity to strengthen the paper.

---

### Meta-Review · Area_Chair_ZyXv · 2025-12-30

**Summary:**

This paper introduces a benchmark to evaluate the physical correctness of generated videos for Newtonian dynamics by fitting extracted object trajectories to physics-informed neural networks (PINN).

Reviewers commented positively on the writing quality, the real-world physical experiments, systematic analyses of prompts and conditioning, and the quantitative and physics-informed evaluation framework.

Reviewers commented negatively on the limited scenarios, strong assumptions, ambiguous 2D trajectories, the comparison with related work, the reliance on SAM and DepthAnything, the limited inclusion of video generation models, the lack of correlation with human judgment, and the fact that the design of invariants and ODE priors is manually specified. These concerns have been partially addressed in the rebuttal.

Overall, reviewers share the same core concerns, such as the limited scenarios and strong assumptions, which may limit the impact of this contribution as a benchmark for future works.

**Reviewer Concerns:**

(Weaknesses are indexed using reviewers' original ordering)

For Reviewer AAtS, W1 has been responded to but with limited evidence. W2 has been partially addressed. W3 has been addressed.

For Reviewer Z6tA, W1 has been responded to but with limited evidence. W2 has been partially addressed. W3, W4, and W5 has been addressed.

For Reviewer MqUN, W1 and W4 have been responded to but with limited evidence. W2 and W3 have been addressed.

For Reviewer LyJG, W1 and W3 have been responded to but with limited evidence. W2 has been partially addressed.  W4 has been addressed.

**Reviewer Scores:**

For Reviewer AAtS, the score is unlikely to be increased.

For Reviewer Z6tA, the score is likely to be the same or slightly increased.

For Reviewer MqUN, the score is likely to be the same.

For Reviewer LyJG, the score is likely to be the same.

---

### Decision · Program_Chairs · 2026-01-26

Reject